# A Restoration Network as an Implicit Prior

**Yuyang Hu**[1], **Mauricio Delbracio**[2], **Peyman Milanfar**[2], **Ulugbek S. Kamilov**[1,2]
[1]Washington University in St. Louis, [2]Google Research
{h.yuyang, kamilov}@wustl.edu,
{mdelbra,milanfar}@google.com

## Abstract

Image denoisers have been shown to be powerful priors for solving inverse problems in imaging. In this work, we introduce a generalization of these methods that allows any image restoration network to be used as an implicit prior. The proposed method uses priors specified by deep neural networks pre-trained as general restoration operators. The method provides a principled approach for adapting state-of-the-art restoration models for other inverse problems. Our theoretical result analyzes its convergence to a stationary point of a global functional associated with the restoration operator. Numerical results show that the method using a super-resolution prior achieves state-of-the-art performance both quantitatively and qualitatively. Overall, this work offers a step forward for solving inverse problems by enabling the use of powerful pre-trained restoration models as priors.

## 1 Introduction

Many problems in computational imaging, biomedical imaging, and computer vision can be formulated as *inverse problems*, where the goal is to recover a high-quality images from its low-quality observations. Imaging inverse problems are generally ill-posed, thus necessitating the use of prior models on the unknown images for accurate inference. While the literature on prior modeling of images is vast, current methods are primarily based on *deep learning (DL)*, where a deep model is trained to map observations to images (Lucas et al., 2018; McCann et al., 2017; Ongie et al., 2020).

Image denoisers have become popular for specifying image priors for solving inverse problems (Venkatakrishnan et al., 2013; Romano et al., 2017; Kadkhodaie & Simoncelli, 2021; Kamilov et al., 2023). Pre-trained denoisers provide a convenient proxy for image priors that does not require the description of the full density of natural images. The combination of state-of-the-art (SOTA) deep denoisers with measurement models has been shown to be effective in a number of inverse problems, including image super-resolution, deblurring, inpainting, microscopy, and medical imaging (Metzler et al., 2018; Zhang et al., 2017b; Meinhardt et al., 2017; Dong et al., 2019; Zhang et al., 2019; Wei et al., 2020; Zhang et al., 2022) (see also the recent reviews (Ahmad et al., 2020; Kamilov et al., 2023)). This success has led to active research on novel methods based on denoiser priors, their theoretical analyses, statistical interpretations, as well as connections to related approaches such as score matching and diffusion models (Chan et al., 2017; Romano et al., 2017; Buzzard et al., 2018; Reehorst & Schniter, 2019; Sun et al., 2019; Sun et al., 2019; Ryu et al., 2019; Xu et al., 2020; Liu et al., 2021; Cohen et al., 2021a; Hurault et al., 2022a;b; Laumont et al., 2022; Gan et al., 2023).

Despite the rich literature on the topic, the prior work has narrowly focused on leveraging the statistical properties of denoisers. There is little work on extending the formalism and theory to priors specified using other types of image restoration operators, such as, for example, deep image super-resolution models. Such extensions would enable new algorithms that can leverage SOTA pre-trained restoration networks for solving other inverse problems. In this paper, we address this gap by developing the ***Deep Restoration Priors (DRP)*** methodology that provides a principled approach for using restoration operators as priors. We show that when the restoration operator is a *minimum mean-squared error (MMSE)* estimator, DRP can be interpreted as minimizing a composite objective function that includes log of the density of the degraded image as the regularizer. Our interpretation extends the recent formalism based on using MMSE denoisers as priors (Bigdeli et al., 2017; Xu et al., 2020; Kadkhodaie & Simoncelli, 2021; Laumont et al., 2022; Gan et al., 2023). We present a theoretical convergence analysis of DRP to a stationary point of the objective function

under a set of clearly specified assumptions. We show the practical relevance of DRP by solving several inverse problems by using a super-resolution network as a prior. Our numerical results show the potential of DRP to adapt the super-resolution model to act as an effective prior that can outperform image denoisers. This work thus addresses a gap in the current literature by providing a new principled framework for using pre-trained restoration models as priors for inverse problems.

All proofs and some details that have been omitted for space appear in the supplementary material.

## 2 BACKGROUND

**Inverse Problems.** Many imaging problems can be formulated as inverse problems that seek to recover an unknown image $\boldsymbol{x} \in \mathbb{R}^n$ from from its corrupted observation

$$\boldsymbol{y} = \boldsymbol{A}\boldsymbol{x} + \boldsymbol{e}, \tag{1}$$

where $\boldsymbol{A} \in \mathbb{R}^{m \times n}$ is a measurement operator and $\boldsymbol{e} \in \mathbb{R}^m$ is the noise. A common strategy for addressing inverse problems involves formulating them as an optimization problem

$$\widehat{\boldsymbol{x}} \in \underset{\boldsymbol{x} \in \mathbb{R}^n}{\arg\min}\, f(\boldsymbol{x}) \quad \text{with} \quad f(\boldsymbol{x}) = g(\boldsymbol{x}) + h(\boldsymbol{x}) , \tag{2}$$

where $g$ is the data-fidelity term that measures the fidelity to the observation $\boldsymbol{y}$ and $h$ is the regularizer that incorporates prior knowledge on $\boldsymbol{x}$. For example, common functionals in imaging inverse problems are the least-squares data-fidelity term $g(\boldsymbol{x}) = \frac{1}{2} \|\boldsymbol{A}\boldsymbol{x} - \boldsymbol{y}\|_2^2$ and the total variation (TV) regularizer $h(\boldsymbol{x}) = \tau \|\boldsymbol{D}\boldsymbol{x}\|_1$, where $\boldsymbol{D}$ is the image gradient, and $\tau > 0$ a regularization parameter.

**Deep Learning.** DL is extensively used for solving imaging inverse problems (McCann et al., 2017; Lucas et al., 2018; Ongie et al., 2020). Instead of explicitly defining a regularizer, DL methods often train convolutional neural networks (CNNs) to map the observations to the desired images (Wang et al., 2016; Jin et al., 2017; Kang et al., 2017; Chen et al., 2017; Delbracio et al., 2021; Delbracio & Milanfar, 2023). Model-based DL (MBDL) is a widely-used sub-family of DL algorithms that integrate physical measurement models with priors specified using CNNs (see reviews (Ongie et al., 2020; Monga et al., 2021)). The literature of MBDL is vast, but some well-known examples include plug-and-play priors (PnP), regularization by denoising (RED), deep unfolding (DU), compressed sensing using generative models (CSGM), and deep equilibrium models (DEQ) (Bora et al., 2017; Romano et al., 2017; Zhang & Ghanem, 2018; Hauptmann et al., 2018; Gilton et al., 2021; Liu et al., 2022). These approaches come with different trade-offs in terms of imaging performance, computational and memory complexity, flexibility, need for supervision, and theoretical understanding.

**Denoisers as Priors.** PnP (Venkatakrishnan et al., 2013; Sreehari et al., 2016) is one of the most popular MBDL approaches for inverse problems based on using deep denoisers as imaging priors (see recent reviews (Ahmad et al., 2020; Kamilov et al., 2023)). For example, the proximal-gradient method variant of PnP can be written as (Hurault et al., 2022a)

$$\boldsymbol{x}^k \leftarrow \mathsf{prox}_{\gamma g}(\boldsymbol{z}^k) \quad \text{with} \quad \boldsymbol{z}^k \leftarrow \boldsymbol{x}^{k-1} - \gamma\tau(\boldsymbol{x}^{k-1} - \mathsf{D}_\sigma(\boldsymbol{x}^{k-1})), \tag{3}$$

where $\mathsf{D}_\sigma$ is a denoiser with a parameter $\sigma > 0$ for controlling its strength, $\tau > 0$ is a regularization parameter, and $\gamma > 0$ is a step-size. The theoretical convergence of PnP methods has been established for convex functions $g$ using monotone operator theory (Sreehari et al., 2016; Sun et al., 2019; Ryu et al., 2019), as well as for nonconvex functions based on interpreting the denoiser as a MMSE estimator (Xu et al., 2020) or ensuring that the term $(\mathsf{I} - \mathsf{D}_\sigma)$ in (3) corresponds to a gradient $\nabla h$ of a function $h$ parameterized by a deep neural network (Hurault et al., 2022a;b; Cohen et al., 2021a). Many variants of PnP have been developed over the past few years (Romano et al., 2017; Metzler et al., 2018; Zhang et al., 2017b; Meinhardt et al., 2017; Dong et al., 2019; Zhang et al., 2019; Wei et al., 2020), which has motivated an extensive research on its theoretical properties (Chan et al., 2017; Buzzard et al., 2018; Ryu et al., 2019; Sun et al., 2019; Tirer & Giryes, 2019; Teodoro et al., 2019; Xu et al., 2020; Sun et al., 2021; Cohen et al., 2021b; Hurault et al., 2022a; Laumont et al., 2022; Hurault et al., 2022b; Gan et al., 2023).

This work is most related to two recent PnP-inspired methods using restoration operators instead of denoisers (Zhang et al., 2019; Liu et al., 2020). Deep plug-and-play super-resolution (DPSR) (Zhang et al., 2019) was proposed to perform image super-resolution under arbitrary blur kernels by using a bicubic super-resolver as a prior. Regularization by artifact removal (RARE) (Liu et al., 2020)

was proposed to use CNNs pre-trained directly on subsampled and noisy Fourier data as priors for magnetic resonance imaging (MRI). These prior methods did not leverage statistical interpretations of the restoration operators to provide a theoretical analysis for the corresponding PnP variants.

It is also worth highlighting the work of Gribonval and colleagues on theoretically exploring the relationship between MMSE restoration operators and proximal operators (Gribonval, 2011; Gribonval & Machart, 2013; Gribonval & Nikolova, 2021). Some of the observations and intuition in that prior line of work is useful for the theoretical analysis of the proposed DRP methodology.

**Our contribution.** *(1)* Our first contribution is the new method DRP for solving inverse problems using the prior implicit in a pre-trained deep restoration network. Our method is a major extension of recent methods (Bigdeli et al., 2017; Xu et al., 2020; Kadkhodaie & Simoncelli, 2021; Gan et al., 2023) from denoisers to more general restoration operators.*(2)* Our second contribution is a new theory that characterizes the solution and convergence of DRP under priors associated with the MMSE restoration operators. Our theory is general in the sense that it allows for nonsmooth data-fidelity terms and expansive restoration models. *(3)* Our third contribution is the implementation of DRP using the popular SwinIR (Liang et al., 2021) super-resolution model as a prior for two distinct inverse problems, namely deblurring and super-resolution. We publicly share our implementation that shows the potential of using restoration models to achieve SOTA performance.

## 3 DEEP RESTORATION PRIOR

Image denoisers are currently extensively used as priors for solving inverse problems. We extend this approach by proposing the following method that uses a more general restoration operator.

---

**Algorithm 1** Deep Restoration Priors (DRP)

---

1: **input:** Initial value $\boldsymbol{x}^0 \in \mathbb{R}^n$ and parameters $\gamma, \tau > 0$
2: **for** $k = 1, 2, 3, \ldots$ **do**
3:     $\boldsymbol{z}^k \leftarrow \boldsymbol{x}^{k-1} - \gamma\tau\mathsf{G}(\boldsymbol{x}^{k-1})$ where $\mathsf{G}(\boldsymbol{x}) := \boldsymbol{x} - \mathsf{R}(\mathbf{H}\boldsymbol{x})$
4:     $\boldsymbol{x}^k \leftarrow \mathsf{sprox}_{\gamma g}(\boldsymbol{z}^k)$
5: **end for**

---

The prior in Algorithm 1 is implemented in Line 3 using a deep model $\mathsf{R} : \mathbb{R}^p \to \mathbb{R}^n$ pre-trained to solve the following restoration problem

$$\boldsymbol{s} = \mathbf{H}\boldsymbol{x} + \boldsymbol{n} \quad \text{with} \quad \boldsymbol{x} \sim p_{\boldsymbol{x}}, \quad \boldsymbol{n} \sim \mathcal{N}(0, \sigma^2\mathbf{I}), \tag{4}$$

where $\mathbf{H} \in \mathbb{R}^{p \times n}$ is a degradation operator, such as blur or downscaling, and $\boldsymbol{n} \in \mathbb{R}^p$ is the *additive white Gaussian noise (AWGN)* of variance $\sigma^2$. The density $p_{\boldsymbol{x}}$ is the prior distribution of the desired class of images. Note that the restoration problem (4) is only used for training $\mathsf{R}$ and doesn't have to correspond to the inverse problem in (1) we are seeking to solve. When $\mathbf{H} = \mathbf{I}$, the restoration operator $\mathsf{R}$ reduces to an AWGN denoiser used in the traditional PnP methods (Romano et al., 2017; Kadkhodaie & Simoncelli, 2021; Hurault et al., 2022a). The goal of DRP is to leverage a pre-trained restoration network $\mathsf{R}$ to gain access to the prior.

The measurement consistency is implemented in Line 4 using the *scaled* proximal operator

$$\mathsf{sprox}_{\gamma g}(\boldsymbol{z}) := \mathsf{prox}_{\gamma g}^{\mathbf{H}^{\mathsf{T}}\mathbf{H}}(\boldsymbol{z}) = \underset{\boldsymbol{x} \in \mathbb{R}^n}{\arg\min} \left\{ \frac{1}{2}\|\boldsymbol{x} - \boldsymbol{z}\|_{\mathbf{H}^{\mathsf{T}}\mathbf{H}}^2 + \gamma g(\boldsymbol{x}) \right\}, \tag{5}$$

where $\|\boldsymbol{v}\|_{\mathbf{H}^{\mathsf{T}}\mathbf{H}} := \boldsymbol{v}^{\mathsf{T}}\mathbf{H}^{\mathsf{T}}\mathbf{H}\boldsymbol{v}$ denotes the weighted Euclidean seminorm of a vector $\boldsymbol{v}$. When $\mathbf{H}^{\mathsf{T}}\mathbf{H}$ is positive definite and $g$ is convex, the functional being minimized in (5) is strictly convex, which directly implies that the solution is unique. On the other hand, when $g$ is not convex or $\mathbf{H}^{\mathsf{T}}\mathbf{H}$ is positive semidefinite, there might be multiple solutions and the scaled proximal operator simply returns one of the solutions. It is also worth noting that (5) has an efficient solution when $g$ is the least-squares data-fidelity term (see for example the discussion in (Kamilov et al., 2023) on efficient implementations of proximal operators of least-squares).

The fixed points of Algorithm 1 can be characterized for subdifferentiable $g$ (see Chapter 3 in (Beck, 2017) for a discussion on subdifferentiability). When DRP converges, it converges to vectors $\boldsymbol{x}^* \in$

$\mathbb{R}^n$ that satisfy (see formal analysis in Supplement A.1)

$$\mathbf{0} \in \partial g(\boldsymbol{x}^*) + \tau \mathbf{H}^\mathsf{T} \mathbf{H} \mathsf{G}(\boldsymbol{x}^*) \tag{6}$$

where $\partial g$ is the subdifferential of $g$ and $\mathsf{G}$ is defined in Line 3 of Algorithm 1. As discussed in the next section, under additional assumptions, one can associate the fixed points of DRP with the stationary points of a composite objective function $f = g + h$ for some regularizer $h$.

## 4  CONVERGENCE ANALYSIS OF DRP

In this section, we present a theoretical analysis of DRP. We first provide a more insightful interpretation of its solutions for restoration models that compute MMSE estimators of (4). We then discuss the convergence of the iterates generated by DRP. Our analysis will require several assumptions.

We will consider restoration models that perform MMSE estimation of $\boldsymbol{x} \in \mathbb{R}^n$ for the problem (4)

$$\mathsf{R}(\boldsymbol{s}) = \mathbb{E}\left[\boldsymbol{x}|\boldsymbol{s}\right] = \int \boldsymbol{x} p_{\boldsymbol{x}|\boldsymbol{s}}(\boldsymbol{x}; \boldsymbol{s})\, \mathsf{d}\boldsymbol{x} = \int \boldsymbol{x} \frac{p_{\boldsymbol{s}|\boldsymbol{x}}(\boldsymbol{s}; \boldsymbol{x}) p_{\boldsymbol{x}}(\boldsymbol{x})}{p_{\boldsymbol{s}}(\boldsymbol{s})}\, \mathsf{d}\boldsymbol{x}. \tag{7}$$

where we used the probability density of the observation $\boldsymbol{s} \in \mathbb{R}^p$

$$p_{\boldsymbol{s}}(\boldsymbol{s}) = \int p_{\boldsymbol{s}|\boldsymbol{x}}(\boldsymbol{s}; \boldsymbol{x}) p_{\boldsymbol{x}}(\boldsymbol{x})\, \mathsf{d}\boldsymbol{x} = \int G_\sigma(\boldsymbol{s} - \mathbf{H}\boldsymbol{x}) p_{\boldsymbol{x}}(\boldsymbol{x})\, \mathsf{d}\boldsymbol{x}. \tag{8}$$

The function $G_\sigma$ in (8) denotes the Gaussian density function with the standard deviation $\sigma > 0$.

**Assumption 1.** *The prior density $p_{\boldsymbol{x}}$ is non-degenerate over $\mathbb{R}^n$.*

For example, a probability density $p_{\boldsymbol{x}}$ is degenerate over $\mathbb{R}^n$, if it is supported on a space of lower dimensions than $n$. Our goal is to establish an explicit link between the MMSE restoration operator (7) and the following regularizer

$$h(\boldsymbol{x}) = -\tau \sigma^2 \log p_{\boldsymbol{s}}(\mathbf{H}\boldsymbol{x}), \quad \boldsymbol{x} \in \mathbb{R}^n, \tag{9}$$

where $\tau$ is the parameter in Algorithm 1, $p_{\boldsymbol{s}}$ is the density of the observation (8), and $\sigma^2$ is the AWGN level used for training the restoration network. We adopt Assumption 1 to have a more intuitive mathematical exposition, but one can in principle generalize the link between MMSE operators and regularization beyond non-degenerate priors (Gribonval & Machart, 2013). It is also worth observing that the function $h$ is infinitely continuously differentiable, since it is obtained by integrating $p_{\boldsymbol{x}}$ with a Gaussian density $G_\sigma$ (Gribonval, 2011; Gribonval & Machart, 2013).

**Assumption 2.** *The scaled proximal operator $\mathsf{sprox}_{\gamma g}$ is well-defined in the sense that there exists a solution to the problem (5) for any $\boldsymbol{z} \in \mathbb{R}^n$. The function $g$ is subdifferentiable over $\mathbb{R}^n$.*

This mild assumption is needed for us to be able to run our method. There are multiple ways to ensure that the scaled proximal operator is well defined. For example, $\mathsf{sprox}_{\gamma g}$ is always well-defined for any $g$ that is proper, closed, and convex (Parikh & Boyd, 2014). This directly makes DRP applicable with the popular least-squares data-fidelity term $g(\boldsymbol{x}) = \frac{1}{2}\|\boldsymbol{y} - \boldsymbol{A}\boldsymbol{x}\|_2^2$. One can relax the assumption of convexity by considering $g$ that is proper, closed, and coercive, in which case $\mathsf{sprox}_{\gamma g}$ will have a solution (see for example Chapter 6 of (Beck, 2017)). Note that we do not require the solution to (5) to be unique; it is sufficient for $\mathsf{sprox}_{\gamma g}$ to return one of the solutions.

We are now ready to theoretically characterize the solutions of DRP.

**Theorem 1.** *Let $\mathsf{R}$ be the MMSE restoration operator (7) corresponding to the restoration problem (4) under Assumptions 1-3. Then, any fixed-point $\boldsymbol{x}^* \in \mathbb{R}^n$ of DRP satisfies*

$$\mathbf{0} \in \partial g(\boldsymbol{x}^*) + \nabla h(\boldsymbol{x}^*),$$

*where $h$ is given in (9).*

The proof of the theorem is provided in the supplement and generalizes the well-known *Tweedie's formula* (Robbins, 1956; Miyasawa, 1961; Gribonval, 2011) to restoration operators. The theorem implies that the solutions of DRP satisfy the first-order conditions for the objective function $f = g + h$. If $g$ is a negative log-likelihood $p_{\boldsymbol{y}|\boldsymbol{x}}$, then the fixed-points of DRP can be interpreted as *maximum-a-posteriori probability (MAP)* solutions corresponding to the prior density $p_{\boldsymbol{s}}$. The density $p_{\boldsymbol{s}}$ is

related to the true prior $p_{\boldsymbol{x}}$ through eq. (8), which implies that DRP has access to the prior $p_{\boldsymbol{x}}$ through the restoration operator R via density $p_{\boldsymbol{s}}$. As $\mathbf{H} \to \mathbf{I}$ and $\sigma \to 0$, the density $p_{\boldsymbol{s}}$ approaches the prior distribution $p_{\boldsymbol{x}}$.

The convergence analysis of DRP will require additional assumptions.

**Assumption 3.** *The data-fidelity term $g$ and the implicit regularizer $h$ are bounded from below.*

This assumption implies that there exists $f^* > -\infty$ such that $f(\boldsymbol{x}) \geq f^*$ for all $\boldsymbol{x} \in \mathbb{R}^n$.

**Assumption 4.** *The function $h$ has a Lipschitz continutous gradient with constant $L > 0$. The degradation operator associated with the restoration network is such that $\lambda \succeq \mathbf{H}^{\mathsf{T}}\mathbf{H} \succeq \mu > 0$.*

This assumption is related to the implicit prior associated with a restoration model and is needed to ensure the monotonic reduction of the objective $f$ by the DRP iterates. As stated under eq. (9), the function $h$ is infinitely continuously differentiable. We additionally adopt the standard optimization assumption that $\nabla h$ is Lipschitz continuous (Nesterov, 2004). It is also worth noting that the positive definiteness of $\mathbf{H}^{\mathsf{T}}\mathbf{H}$ in Assumption 4 is a relaxation of the traditional PnP assumption that the prior is a denoiser, which makes our theoretical analysis a significant extension of the prior work (Bigdeli et al., 2017; Xu et al., 2020; Kadkhodaie & Simoncelli, 2021; Gan et al., 2023).

We are now ready to state the following results.

**Theorem 2.** *Run DRP for for $t \geq 1$ iterations under Assumptions 1-4 using a step-size $\gamma = \mu/(\alpha L)$ with $\alpha > 1$. Then, for each iteration $1 \leq k \leq t$, there exists $\boldsymbol{w}(\boldsymbol{x}^k) \in \partial f(\boldsymbol{x}^k)$ such that*

$$\frac{1}{t} \sum_{k=1}^{t} \|\boldsymbol{w}(\boldsymbol{x}^k)\|_2^2 \leq \frac{C(f(\boldsymbol{x}^0) - f^*)}{t},$$

*where $C > 0$ is an iteration independent constant.*

The exact expression for the constant $C$ is given in the proof. Theorem 2 shows that the iterates generated by DRP satisfy $\boldsymbol{w}(\boldsymbol{x}^k) \to \boldsymbol{0}$ as $t \to \infty$. Theorems 1 and 2 do not explicitly require convexity or smoothness of $g$, and non-expansiveness of R. They can thus be viewed as a major generalization of the existing theory from denoisers to more general restoration operators.

## 5 NUMERICAL RESULTS

We now numerically validate DRP on several distinct inverse problems. Due to space limitations in the main paper, we have included several additional numerical results in the supplementary material.

We consider two inverse problems of form $\boldsymbol{y} = \boldsymbol{A}\boldsymbol{x} + \boldsymbol{e}$: (a) *Image Deblurring* and (b) *Single Image Super Resolution (SISR)*. For both problems, we assume that $\boldsymbol{e}$ is the additive white Gaussian noise (AWGN). We adopt the traditional $\ell_2$-norm loss as the data-fidelity term in (2) for both problems. We use the Peak Signal-to-Noise Ratio (PSNR) for quantitative performance evaluation.

In the main manuscript, we compare DRP with several variants of denoiser-based methods, including SD-RED (Romano et al., 2017), PnP-ADMM (Chan et al., 2017), IRCNN (Zhang et al., 2017b), and DPIR (Zhang et al., 2022). SD-RED and PnP-ADMM refer to the steepest-descent variant of RED and the ADMM variant of PnP, both of which incorporate AWGN denoisers based on DnCNN (Zhang et al., 2017a). IRCNN and DPIR are based on half-quadratic splitting (HQS) iterations that use the IRCNN and the DRUNet denoisers, respectively.

In the supplement, we present several additional comparisons, namely: (a) evaluation of the performance of DRP on the task of image denoising; (b) additional comparison of DRP with the recent provably convergent variant of PnP called gradient-step plug-and-play (GS-PnP) (Hurault et al., 2022a); (c) comparison of DRP with the diffusion posterior sampling (DPS) (Chung et al., 2023) method that uses a denoising diffusion model as a prior; (d) illustration of the improvement of DRP using SwinIR as a prior over the direct application of SwinIR on SR using the Gaussian kernel; (e) presentation of quantitative results using SSIM and LPIPS metrics; (f) additional comparison with DPIR using with SwinIR trained as denoiser; (g) additional evaluation of DRP with a non-MMSE restoration priors and other MMSE restoration priors; and (h) evaluation of robustness of our numerical results to random seed.

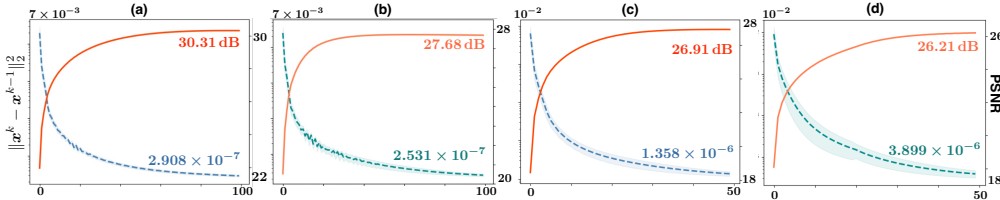

Figure 1: Illustration of the convergence behaviour of DRP for image deblurring and single image super resolution on the Set3c dataset. *(a)-(b)*: Deblurring with Gaussian blur kernels of standard deviations 1.6 and 2.0. *(c)-(d)*: $2\times$ and $3\times$ super resolution with the Gaussian blur kernel of standard deviation 2.0. Average distance $\|\boldsymbol{x}^k - \boldsymbol{x}^{k-1}\|_2^2$ and PSNR relative to the groundtruth are plotted, with shaded areas indicating the standard deviation of these metrics across all test images.

| Kernel | Datasets | SD-RED | PnP-ADMM | IRCNN+ | DPIR | DRP |
|---|---|---|---|---|---|---|
| | Set3c | 27.14/0.121 | 29.11/0.099 | 28.14/0.086 | 29.53/0.110 | **30.69/0.056** |
| | Set5 | 29.78/0.160 | 32.31/0.137 | 29.46/0.130 | 32.38/0.152 | **32.79/0.109** |
| | CBSD68 | 25.78/0.344 | 28.90/0.287 | 26.86/0.307 | 28.86/0.307 | **29.10/0.246** |
| | McMaster | 29.69/0.167 | 32.20/0.136 | 29.15/0.153 | 32.42/0.151 | **32.79/0.107** |
| | Set3c | 25.83/0.165 | 27.05/0.157 | 26.58/0.156 | 27.52/0.159 | **27.89/0.088** |
| | Set5 | 28.13/0.193 | 30.77/0.181 | 28.75/0.182 | 30.94/0.188 | **31.04/0.143** |
| | CBSD68 | 24.43/0.455 | 27.45/0.360 | 25.97/0.416 | **27.52**/0.376 | 27.46/**0.296** |
| | McMaster | 28.71/0.201 | 30.50/0.185 | 28.27/0.222 | 30.78/0.193 | **30.79/0.141** |

Table 1: PSNR↑/LPIPS↓ comparison of DRP and several SOTA methods for solving inverse problems using denoisers on image deblurring with the Gaussian blur kernels of standard deviation 1.6 and 2.0 on Set3c, Set5, CBSD68 and McMaster datasets. The **best** and second best results are highlighted. Note how DRP can outperform SOTA PnP methods that use denoisers as priors.

## 5.1 SWIN TRANSFORMER BASED SUPER RESOLUTION PRIOR

**Super Resolution Network Architecture.** We pre-trained a $q\times$ super resolution model $\mathsf{R}_q$ using the SwinIR (Liang et al., 2021) architecture based on Swin Transformer. Our training dataset comprised both the DIV2K (Agustsson & Timofte, 2017) and Flick2K (Lim et al., 2017) dataset, containing 3450 color images in total. During training, we applied $q\times$ bicubic downsampling to the input images with AWGN characterized by standard deviation $\sigma$ randomly chosen in [0, 10/255]. We used three SwinIR SR models, each trained for different down-sampling factors: $2\times$, $3\times$ and $4\times$.

**Prior Refinement Strategy for the Super Resolution prior.** Theorem 1 suggests that as $\mathbf{H} \rightarrow \mathbf{I}$, the prior in DRP converges to $p_{\boldsymbol{x}}$. This process can be approximated for SwinIR by controlling the down-sampling factor $q$ of the SR restoration prior $\mathsf{R}_q(\cdot)$. We observed through our numerical experiments that gradual reduction of $q$ leads to less reconstruction artifacts and enhanced fine details. We will denote the approach of gradually reducing $q$ as *prior refinement strategy*. We initially set $q$ to a larger down-sampling factor, which acts as a more aggressive prior; we then reduce $q$ to a smaller value leading to preservation of finer details. This strategy is conceptually analogous to the gradual reduction of $\sigma$ in the denoiser in the SOTA PnP methods such as DPIR (Zhang et al., 2022).

## 5.2 IMAGE DEBLURRING

Image deblurring is based on the degradation operator of the form $\boldsymbol{A} = \boldsymbol{K}$, where $\boldsymbol{K}$ is a convolution with the blur kernel $\boldsymbol{k}$. We consider image deblurring using two $25 \times 25$ Gaussian kernels (with the standard deviations 1.6 and 2) used in (Zhang et al., 2019), and the AWGN vector $\boldsymbol{e}$ corresponding to noise level of 2.55/255. For fair comparison, we use the official implementations provided by each baseline method and use the same random seed to ensure consistency of random noise for all methods. The restoration model used as a prior in DRP is SwinIR introduced in Section 5.1, so that the operation $\mathbf{H}$ corresponds to the standard bicubic downsampling. The scaled proximal operator

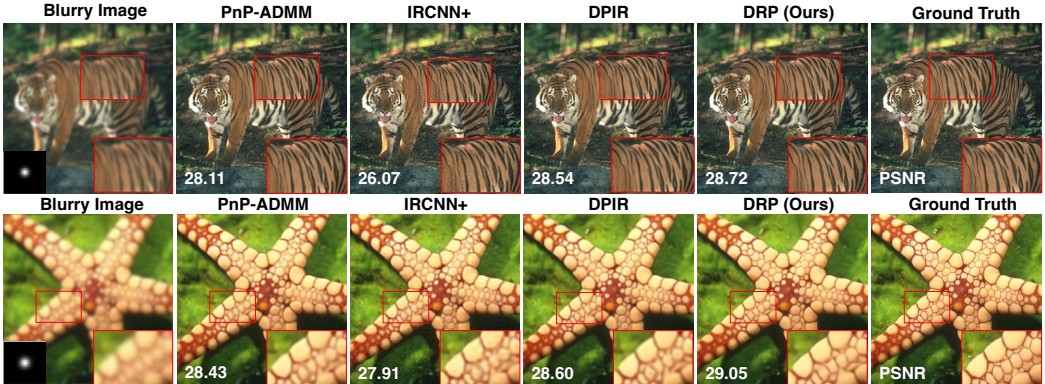

Figure 2: Visual comparison of DRP with several well-known methods on Gaussian deblurring of color images. The top row shows results for a blur kernel with a standard deviation (std) of 1.6, while the bottom row shows results for another blur kernel with std = 2. The squares at the bottom-left corner of blurry images show the blur kernels. Each image is labeled by its PSNR in dB with respect to the original image. The visual differences are highlighted in the bottom-right corner. Note how DRP using restoration prior improves over SOTA methods based on denoiser priors.

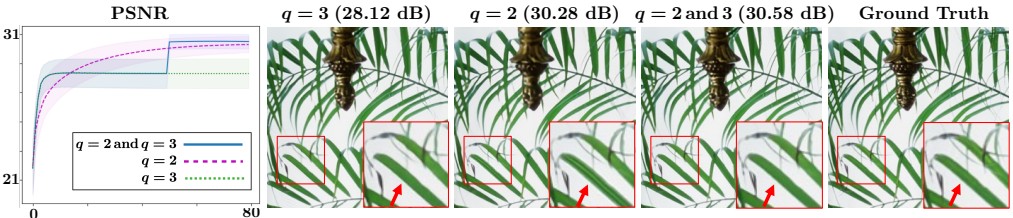

Figure 3: Illustration of the impact of different SR factors in the prior used within DRP for image deblurring. We show three scenarios: (i) using only $3\times$ prior, (ii) using only $2\times$ prior, and (iii) the use of the *prior refinement strategy*, which combines both the $2\times$ and $3\times$ priors. ***Left***: Convergence of PSNR against the iteration number for all three configurations. ***Right***: Visual illustration of the final image for each setting. The visual difference is highlighted by the red arrow in the zoom-in box. Note how the reduction of $q$ can lead to about 0.3 dB improvement in the final performance.

$\mathsf{sprox}_{\lambda g}$ in (5) with data-fidelity term $g(\boldsymbol{x}) = \frac{1}{2}\|\boldsymbol{y} - \boldsymbol{K}\boldsymbol{x}\|_2^2$ can be written as

$$\mathsf{sprox}_{\gamma g}(\boldsymbol{z}) = (\boldsymbol{K}^\mathsf{T}\boldsymbol{K} + \gamma \mathbf{H}^\mathsf{T}\mathbf{H})^{-1}[\boldsymbol{K}^\mathsf{T}\boldsymbol{y} + \gamma \mathbf{H}^\mathsf{T}\mathbf{H}\boldsymbol{z}]. \tag{10}$$

We adopt a standard approach of using a few iterations of the conjugate gradient (CG) method (see for example (Aggarwal et al., 2019)) to implement the scaled proximal operator (10) by avoiding the direct inversion of $(\boldsymbol{K}^\mathsf{T}\boldsymbol{K} + \gamma \mathbf{H}^\mathsf{T}\mathbf{H})$. As bicubic SR model is adopted as prior, the $\mathsf{R}(\mathbf{H}\boldsymbol{x})$ in Step 3 of Algorithm 1 performs a bicubic downsampling of the intermediate image $\boldsymbol{x}^{k-1}$ and inputs it into the bicubic SR SwinIR. In each DRP iteration, we run three steps of a CG solver, starting from a warm initialization from the previous DRP iteration. We fine-turned the hyper-parameter $\gamma$, $\tau$ and SR restoration prior rate $q$ to achieve the highest PSNR value on the Set5 dataset and then apply the same configuration to the other three datasets.

Figure 1 (a)-(b) illustrates the convergence behaviour of DRP on the Set3c dataset for two blur kernels. Table 1 presents the quantitative evaluation of the reconstruction performance on two different blur kernels, showing that DRP outperforms the baseline methods across four widely-used datasets. Figure 2 visually illustrates the reconstructed results on the same two blur kernels. Note how DRP can reconstruct the fine details of the tiger and starfish, as highlighted within the zoom-in boxes, while all the other baseline methods yield either oversmoothed reconstructions or noticeable artifacts. These results show that DRP can leverage SwinIR as an implicit prior, which not only ensures stable convergence, but also leads to competitive performance when compared to denoisers priors.

| SR | Kernel | Datasets | SD-RED | PnP-ADMM | IRCNN+ | DPIR | DRP |
|---|---|---|---|---|---|---|---|
| 2× | | Set3c | 27.01/0.147 | 27.88/0.131 | 27.48/0.113 | 28.18/0.137 | **29.26/0.079** |
| | | Set5 | 28.98/0.207 | 31.41/0.171 | 29.47/0.194 | 31.42/0.175 | **31.47/0.120** |
| | | CBSD68 | 26.11/0.426 | 28.00/0.338 | 26.66/0.413 | 27.97/0.356 | **28.12/0.285** |
| | | McMaster | 28.70/0.257 | 30.98/0.208 | 29.11/0.237 | 31.16/0.179 | **31.39/0.122** |
| | | Set3c | 25.20/0.191 | 25.86/0.187 | 25.92/0.171 | 26.80/0.171 | **27.41/0.107** |
| | | Set5 | 28.57/0.246 | 30.06/0.205 | 28.91/0.241 | 30.36/0.241 | **30.42/0.163** |
| | | CBSD68 | 25.77/0.529 | 26.88/0.393 | 26.06/0.510 | 26.98/0.401 | **26.98/0.327** |
| | | McMaster | 28.15/0.303 | 29.53/0.219 | 28.41/0.299 | 29.87/0.215 | **30.03/0.167** |
| 3× | | Set3c | 25.50/0.177 | 25.85/0.173 | 25.72/0.158 | 26.64/0.169 | **27.77/0.090** |
| | | Set5 | 28.75/0.220 | 30.09/0.195 | 29.14/0.216 | 30.39/0.202 | **30.83/0.149** |
| | | CBSD68 | 25.69/0.475 | 26.78/0.378 | 26.01/0.460 | 26.80/0.394 | **27.18/0.299** |
| | | McMaster | 28.38/0.269 | 29.52/0.208 | 28.53/0.262 | 29.82/0.210 | **29.92/0.164** |
| | | Set3c | 24.55/0.220 | 24.87/0.216 | 24.87/0.196 | 25.84/0.195 | **26.84/0.108** |
| | | Set5 | 28.19/0.280 | 29.26/0.229 | 28.37/0.265 | 29.70/0.225 | **29.88/0.162** |
| | | CBSD68 | 25.40/0.556 | 26.28/0.420 | 25.56/0.530 | 26.39/0.422 | **26.60/0.324** |
| | | McMaster | 27.79/0.329 | 28.72/0.246 | 27.85/0.320 | 29.11/0.236 | **29.47/0.167** |

Table 2: PSNR↑/LPIPS↓ comparison of DRP and several baselines for SISR on Set3c, Set5, CBSD68 and McMaster datasets. The **best** and second best results are highlighted. Note the excellent quantitative performance of DRP, which suggests the potential of using general restoration models as priors.

Figure 3 illustrates the impact of the *prior-refinement strategy* described in Section 5.1. We compare three settings: (i) use of only 3× prior, (ii) use of only 2× prior, and (iii) use of the prior-refinement strategy to leverage both 3× and 2× priors. The subfigure on the left shows the convergence of DRP for each configuration, while the ones on the right show the final imaging quality. Note how the reduction of $q$ leads to better performance, which is analogous to what was observed with the reduction of $\sigma$ in the SOTA PnP methods (Zhang et al., 2022).

## 5.3 SINGLE IMAGE SUPER RESOLUTION

We apply DRP using the bicubic SwinIR prior to Single Image Super Resolution (SISR) task. The measurement operator in SISR can be written as $A = SK$, where $K$ is convolution with the blur kernel $k$ and $S$ performs standard $d$-fold down-sampling with $d^2 = n/m$. The scaled proximal operator $\mathsf{sprox}_{\lambda g}$ in (5) with data-fidelity term $g(x) = \frac{1}{2} \|y - SKx\|_2^2$ can be write as:

$$\mathsf{sprox}_{\gamma g}(z) = (K^\mathsf{T} S^\mathsf{T} SK + \gamma \mathbf{H}^\mathsf{T} \mathbf{H})^{-1}[K^\mathsf{T} S^\mathsf{T} y + \gamma \mathbf{H}^\mathsf{T} \mathbf{H} z], \qquad (11)$$

where $\mathbf{H}$ is the bicubic downsampling operator. Similarly to deblurring in Section 5.2, we use CG to efficiently compute (11). We adjust the hyper-parameter $\gamma$, $\tau$, and the SR restoration prior factor $q$ for the best PSNR performance on Set5, and then use these parameters on the remaining datasets.

We evaluate super-resolution performance across two $25 \times 25$ Gaussian blur kernels, each with distinct standard deviations (1.6 and 2.0), and for two distinct downsampling factors (2× and 3×), incorporating an AWGN vector $e$ corresponding to noise level of 2.55/255. For fair comparison, we use the official implementations provided by each baseline method and use the same random seed to ensure consistency of random noise for all methods.

Figure 1 (c)-(d) illustrates the convergence behaviour of DRP on the Set3c dataset for 2× and 3× SISR. Figure 4 shows the visual reconstruction results for the same downsampling factors. Table 2 summarizes the PSNR values achieved by DRP relative to other baseline methods when applied to different blur kernel and downsampling factors on four commonly used datasets.

It is worth highlighting that the SwinIR model used in DRP was pre-trained for the bicubic super-resolution task. Consequently, the direct application of the pre-trained SwinIR to the setting considered in this section leads to the suboptimal performance due to mismatch between the kernels used. See Supplement B.4 to see how DRP improves over the direct application of SwinIR.

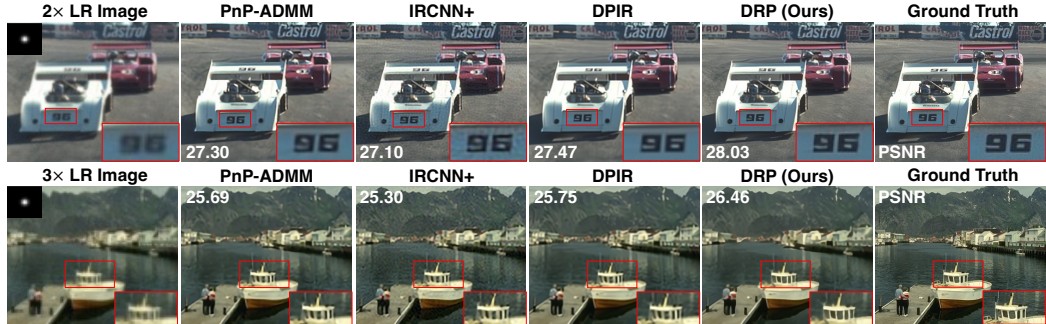

Figure 4: Visual comparison of DRP and several well-known methods on single image super resolution. The top row displays performances for $2\times$ SR, while the bottom row showcases results for $3\times$ SR. The lower-left corner of each low-resolution image shows the blur kernels. Each image is labeled by its PSNR in dB with respect to the original image. The visual differences are highlighted by the boxes in the bottom-right corner. Note the excellent performance of the proposed DRP method using the SwinIR prior both visually and in terms of PSNR.

## 6 CONCLUSION

The work presented in this paper proposes a new DRP method for solving imaging inverse problems by using pre-trained restoration operators as priors, presents its theoretical analysis in terms of convergence, and applies the method to two well-known inverse problems. The proposed method and its theoretical analysis extend the recent work using denoisers as priors by considering more general restoration operators. The numerical validation of DRP shows the improvements due to the use of learned SOTA super-resolution models. One conclusion of this work is the potential effectiveness of going beyond priors specified by traditional denoisers.

### LIMITATIONS AND FUTURE WORK

The work presented in this paper comes with several limitations. The proposed DRP method uses pre-trained restoration models as priors, which means that its performance is inherently limited by the quality of the pre-trained model. As shown in this paper, pre-trained restoration models provide a convenient, principled, and flexible mechanism to specify priors; yet, they are inherently self-supervised and their empirical performance can thus be suboptimal compared to priors trained in a supervised fashion for a specific inverse problem. Our theory is based on the assumption that the restoration prior used for inference performs MMSE estimation. While this assumption is reasonable for deep networks trained using the MSE loss, it is not directly applicable to denoisers trained using other common loss functions, such as the $\ell_1$-norm or SSIM. Finally, as is common with most theoretical work, our theoretical conclusions only hold when our assumptions are satisfied, which might limit their applicability in certain settings.

Our work opens several interesting directions for future research. First, the implicit prior in 9 can be seen as an analysis prior with a transform $\mathbf{H}$ (Elad et al., 2007; Selesnick & Figueiredo, 2009), which suggests a possibility of considering broader class of linear transforms for priors. Second, the excellent performance of the restoration priors beyond denoisers lead to an interesting open question: what is the optimal linear transform $\mathbf{H}$ for a given measurement operator $\boldsymbol{A}$. Third, while in this paper we considered non-blind inverse problems, the extension of DRP to blind inverse problems would be an interesting future direction of research (Gan et al., 2023).

### REPRODUCIBILITY STATEMENT

We have provided the anonymous source code in the supplementary materials. The included README.md file contains detailed instructions on how to run the code and reproduce the results reported in the paper. The pseudo-code of DRP is outlined in Algorithm 1. The complete proofs and technical details for our theoretical analysis can be found in the supplement.

ETHICS STATEMENT

To the best of our knowledge this work does not give rise to any significant ethical concerns.

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

This supplement to the main paper presents the details of our mathematical analysis and provides additional numerical results. The mathematical analysis builds on two distinct lines of work: (a) relationship between the MMSE denoising and the probability density of the noisy image (Gribonval, 2011; Gribonval & Machart, 2013; Bigdeli et al., 2017; Kadkhodaie & Simoncelli, 2021); (b) analysis of the proximal-gradient method (Beck & Teboulle, 2009; Attouch et al., 2013; Parikh & Boyd, 2014; Hurault et al., 2022a). Our results can be viewed as an extension of this line of research to priors specified using more general restoration operators.

The structure of this supplementary document is as follows. In Section A.1, we provide formal characterization of the fixed-points of DRP. In Section A.2, we prove the convergence of DRP under additional assumptions on $g$ and R. In Section B, we provide a number of interesting numerical results that didn't make it to the main paper due to space. In particular, we evaluate the AWGN denoising performance of DRP, compare it with another recent PnP method called GS-PnP, and show how DRP enables adaptation of SwinIR beyond problems it was trained on.

# A THEORETICAL ANALYSIS OF DRP

## A.1 PROOF OF THEOREM 1

**Theorem.** *Let* R *be the MMSE restoration operator* (7) *corresponding to the restoration problem* (4) *under Assumptions 1-3. Then, any fixed-point* $\boldsymbol{x}^* \in \mathbb{R}^n$ *of DRP satisfies*

$$\mathbf{0} \in \partial g(\boldsymbol{x}^*) + \nabla h(\boldsymbol{x}^*),$$

*where* $h$ *is given in* (9).

*Proof.* First note that any fixed point $\boldsymbol{x}^* \in \mathbb{R}^n$ of the DRP method can be expressed as

$$\boldsymbol{x}^* = \mathsf{sprox}_{\gamma g}(\boldsymbol{x}^* - \gamma \tau \mathsf{G}(\boldsymbol{x}^*)) = \arg\min_{\boldsymbol{x} \in \mathbb{R}^n} \left\{ \frac{1}{2} \|\boldsymbol{x} - (\boldsymbol{x}^* - \gamma \tau \mathsf{G}(\boldsymbol{x}^*))\|_{\mathbf{H}^\mathsf{T}\mathbf{H}}^2 + \gamma g(\boldsymbol{x}) \right\}, \quad (12)$$

where we used the definition of the scaled proximal operator. From the optimality conditions of the scaled proximal operator, we then get

$$\mathbf{0} \in \partial g(\boldsymbol{x}^*) + \tau \mathbf{H}^\mathsf{T}\mathbf{H}\mathsf{G}(\boldsymbol{x}^*). \quad (13)$$

On the other hand, the gradient of $p_{\boldsymbol{s}}$, defined in (8), can be expressed as

$$\nabla_{\boldsymbol{s}} p_{\boldsymbol{s}}(\boldsymbol{s}) = \int \left( \frac{1}{\sigma^2} (\mathbf{H}\boldsymbol{x} - \boldsymbol{s}) \right) G_\sigma(\boldsymbol{s} - \mathbf{H}\boldsymbol{x}) p_{\boldsymbol{x}}(\boldsymbol{x}) \, \mathsf{d}\boldsymbol{x} = \frac{1}{\sigma^2} (\mathsf{HR}(\boldsymbol{s}) - \boldsymbol{s} p_{\boldsymbol{s}}(\boldsymbol{s})), \quad (14)$$

where we used the gradient of the Gaussian density with respect to $\boldsymbol{s}$ and the definition of the MMSE restoration operator in eq. (7). By rearranging the terms, we obtain the following relationship

$$\mathbf{H}\mathsf{R}(\boldsymbol{s}) - \boldsymbol{s} = \sigma^2 \nabla \log p_{\boldsymbol{s}}(\boldsymbol{s}), \quad \boldsymbol{s} \in \mathbb{R}^p. \quad (15)$$

By using the definitions $\mathsf{G}(\boldsymbol{x}) = \boldsymbol{x} - \mathsf{R}(\mathbf{H}\boldsymbol{x})$ and $h(\boldsymbol{x}) = -\tau\sigma^2 \log p_{\boldsymbol{s}}(\mathbf{H}\boldsymbol{x})$, with $\boldsymbol{x} \in \mathbb{R}^n$, in (15), we obtain the following generalization of the well-known Tweedie's formula

$$\mathbf{H}^\mathsf{T}\mathbf{H}\mathsf{G}(\boldsymbol{x}) = \mathbf{H}^\mathsf{T}\mathbf{H}(\boldsymbol{x} - \mathsf{R}(\mathbf{H}\boldsymbol{x})) = -\sigma^2 \nabla \log p_{\boldsymbol{s}}(\mathbf{H}\boldsymbol{x}) = \frac{1}{\tau} \nabla h(\boldsymbol{x}). \quad (16)$$

By combining (12) and (16), we directly obtain the desired result

$$\mathbf{0} \in \partial g(\boldsymbol{x}^*) + \nabla h(\boldsymbol{x}^*).$$

$\square$

## A.2 PROOF OF THEOREM 2

**Theorem.** *Run DRP for for $t \geq 1$ iterations under Assumptions 1-4 using a step-size $\gamma = \mu/(\alpha L)$ with $\alpha > 1$. Then, for each iteration $1 \leq k \leq t$, there exists $\boldsymbol{w}(\boldsymbol{x}^k) \in \partial f(\boldsymbol{x}^k)$ such that*

$$\min_{1 \leq k \leq t} \|\boldsymbol{w}(\boldsymbol{x}^k)\|_2^2 \leq \frac{1}{t} \sum_{k=1}^{t} \|\boldsymbol{w}(\boldsymbol{x}^k)\|_2^2 \leq \frac{C(f(\boldsymbol{x}^0) - f^*)}{t},$$

*where $C > 0$ is an iteration independent constant.*

*Proof.* Consider the iteration $k \geq 1$ of DRP

$$\boldsymbol{x}^k = \mathsf{sprox}_{\gamma g}\left(\boldsymbol{x}^{k-1} - \gamma \tau \mathsf{G}(\boldsymbol{x}^{k-1})\right) \quad \text{with} \quad \mathsf{G} := \boldsymbol{x} - \mathsf{R}(\mathbf{H}\boldsymbol{x}),$$

where R is the MMSE restoration operator specified in (7). This implies that $\boldsymbol{x}^k$ minimizes

$$\varphi(\boldsymbol{x}) := \frac{1}{2\gamma}(\boldsymbol{x} - \boldsymbol{x}^{k-1})^{\mathsf{T}}\mathbf{H}^{\mathsf{T}}\mathbf{H}(\boldsymbol{x} - \boldsymbol{x}^{k-1}) + \left[\tau \mathbf{H}^{\mathsf{T}}\mathbf{H}\mathsf{G}(\boldsymbol{x}^{k-1})\right]^{\mathsf{T}}(\boldsymbol{x} - \boldsymbol{x}^{k-1}) + g(\boldsymbol{x})$$

$$= \frac{1}{2\gamma}(\boldsymbol{x} - \boldsymbol{x}^{k-1})^{\mathsf{T}}\mathbf{H}^{\mathsf{T}}\mathbf{H}(\boldsymbol{x} - \boldsymbol{x}^{k-1}) + \nabla h(\boldsymbol{x}^{k-1})^{\mathsf{T}}(\boldsymbol{x} - \boldsymbol{x}^{k-1}) + g(\boldsymbol{x}),$$

where in the second inequality we used eq. (16) from the proof in Supplement A.1. By evaluating $\varphi$ at $\boldsymbol{x}^k$ and $\boldsymbol{x}^{k-1}$, we obtain the following useful inequality

$$g(\boldsymbol{x}^k) \leq g(\boldsymbol{x}^{k-1}) - \frac{1}{2\gamma}(\boldsymbol{x}^k - \boldsymbol{x}^{k-1})^{\mathsf{T}}\mathbf{H}^{\mathsf{T}}\mathbf{H}(\boldsymbol{x}^k - \boldsymbol{x}^{k-1}) - \nabla h(\boldsymbol{x}^{k-1})^{\mathsf{T}}(\boldsymbol{x}^k - \boldsymbol{x}^{k-1}). \quad (17)$$

On the other hand, from the $L$-Lipschitz continuity of $\nabla h$, we have the following bound

$$h(\boldsymbol{x}^k) \leq h(\boldsymbol{x}^{k-1}) + \nabla h(\boldsymbol{x}^{k-1})^{\mathsf{T}}(\boldsymbol{x}^k - \boldsymbol{x}^{k-1}) + \frac{L}{2}\|\boldsymbol{x}^k - \boldsymbol{x}^{k-1}\|_2^2. \quad (18)$$

By combining eqs. (17) and (18), we obtain

$$f(\boldsymbol{x}^k) \leq f(\boldsymbol{x}^{k-1}) - \frac{1}{2}(\boldsymbol{x}^k - \boldsymbol{x}^{k-1})^{\mathsf{T}}\left[\frac{1}{\gamma}\mathbf{H}^{\mathsf{T}}\mathbf{H} - L\mathbf{I}\right](\boldsymbol{x}^k - \boldsymbol{x}^{k-1})$$

$$\leq f(\boldsymbol{x}^{k-1}) - (\alpha - 1)\frac{L}{2}\|\boldsymbol{x}^k - \boldsymbol{x}^{k-1}\|_2^2, \quad (19)$$

where we used $\gamma = \mu/(\alpha L)$ with $\alpha > 1$ and $\mu > 0$ defined in Assumption 4.

On the other hand, from the optimality conditions for $\varphi$, we also have

$$\mathbf{0} \in \mathbf{H}^{\mathsf{T}}\mathbf{H}(\boldsymbol{x}^k - \boldsymbol{x}^{k-1} + \gamma\tau\mathsf{G}(\boldsymbol{x}^{k-1})) + \gamma\partial g(\boldsymbol{x}^k)$$

$$\Leftrightarrow \quad \frac{1}{\gamma}\mathbf{H}^{\mathsf{T}}\mathbf{H}(\boldsymbol{x}^k - \boldsymbol{x}^{k-1}) \in \partial g(\boldsymbol{x}^k) + \nabla h(\boldsymbol{x}^{k-1}),$$

where we used eq. (16) from Supplement A.1. This directly implies that the following inclusion

$$\boldsymbol{w}(\boldsymbol{x}^k) := \frac{1}{\gamma}\mathbf{H}^{\mathsf{T}}\mathbf{H}(\boldsymbol{x}^k - \boldsymbol{x}^{k-1}) + \nabla h(\boldsymbol{x}^k) - \nabla h(\boldsymbol{x}^{k-1}) \in \partial f(\boldsymbol{x}^k)$$

The norm of the subgradient $\boldsymbol{w}(\boldsymbol{x}^k)$ can be bounded as follows

$$\|\boldsymbol{w}(\boldsymbol{x}^k)\|_2 \leq \frac{1}{\gamma}\|\mathbf{H}^{\mathsf{T}}\mathbf{H}(\boldsymbol{x}^k - \boldsymbol{x}^{k-1})\|_2 + \|\nabla h(\boldsymbol{x}^k) - \nabla h(\boldsymbol{x}^{k-1})\|_2$$

$$\leq L\left(\alpha(\lambda/\mu) + 1\right)\|\boldsymbol{x}^k - \boldsymbol{x}^{k-1}\|_2, \quad (20)$$

where we used the Lipschitz constant of $\nabla h$, $\gamma = \mu/(\alpha L)$, and $\lambda \geq \mu > 0$ defined in Assumption 4.

By combining eqs. (19) and (20), we obtain the following inequality

$$\|\boldsymbol{w}(\boldsymbol{x}^k)\|_2^2 \leq A_1\|\boldsymbol{x}^k - \boldsymbol{x}^{k-1}\|_2^2 \leq A_2(f(\boldsymbol{x}^{k-1}) - f(\boldsymbol{x}^k)), \quad (21)$$

where $A_1 := L^2(\alpha(\lambda/\mu) + 1)^2 > 0$ and $A_2 := 2A_1/(L(\alpha - 1)) > 0$. Hence, by averaging over $t \geq 1$ iterations, we can directly get the desired result

$$\min_{1 \leq k \leq t}\|\boldsymbol{w}(\boldsymbol{x}^k)\|_2^2 \leq \frac{1}{t}\sum_{k=1}^{t}\|\boldsymbol{w}(\boldsymbol{x}^k)\|_2^2 \leq \frac{A_2(f(\boldsymbol{x}^0) - f^*)}{t}. \quad (22)$$

This implies that $\boldsymbol{w}(\boldsymbol{x}^k) \to \mathbf{0}$ as $t \to \infty$. $\qquad \square$

## B  ADDITIONAL NUMERICAL RESULTS

### B.1  IMAGE DENOISING

In this subsection, we show the performance of DRP on Gaussian image denoising. The corresponding measurement model is $y = x + e$, where $e$ is AWGN with the standard deviation $\sigma$ and $x$ is the unknown clean image. We use the same SwinIR SR model, introduced in 5.1, as the prior for denoising. The degradation model in the SwinIR prior is the operation $\mathbf{H}$ corresponding to bicubic downsampling. The scaled proximal operator $\mathsf{sprox}_{\lambda g}$ in (5) with data-fidelity term $g(x) = \frac{1}{2}\|y - x\|_2^2$ can be written as

$$\mathsf{sprox}_{\gamma g}(z) := (\mathbf{I} + \gamma \mathbf{H}^\mathsf{T}\mathbf{H})^{-1}[y + \gamma \mathbf{H}^\mathsf{T}\mathbf{H}z], \tag{23}$$

which can be efficiently implemented using CG, as in Section 5.2.

We compare DRP with one of SOTA denoising model DRUNet (Zhang et al., 2019) on noise level ($\sigma = 0.1$). Figure 5 and Figure 6 illustrate the visual performance of DRP on the Set5 and CBSD68 datasets, respectively. Figure 7 further explores the impact of using different SR factors $q$ as priors, elucidating how these choices influence the visual quality of denoising.

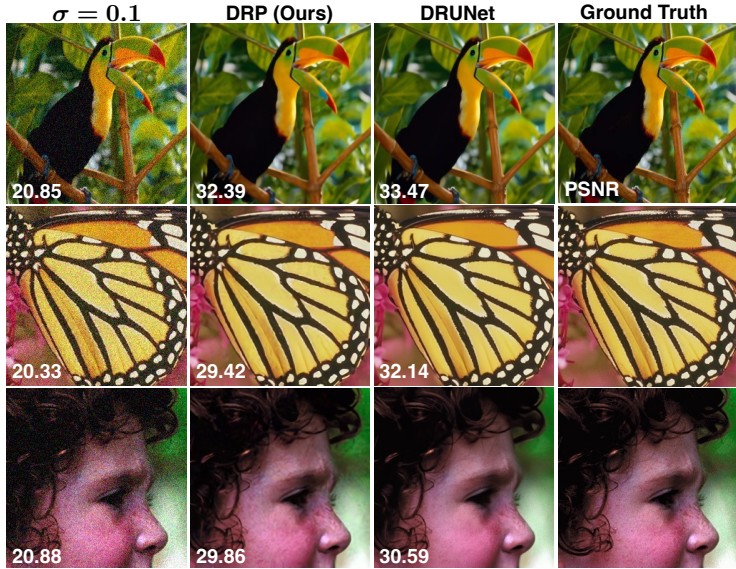

Figure 5: Illustration of denoising results of DRP on Set5 dataset with noise level $\sigma = 0.1$. Each image is labeled by its PSNR (dB) with respect to the original image.

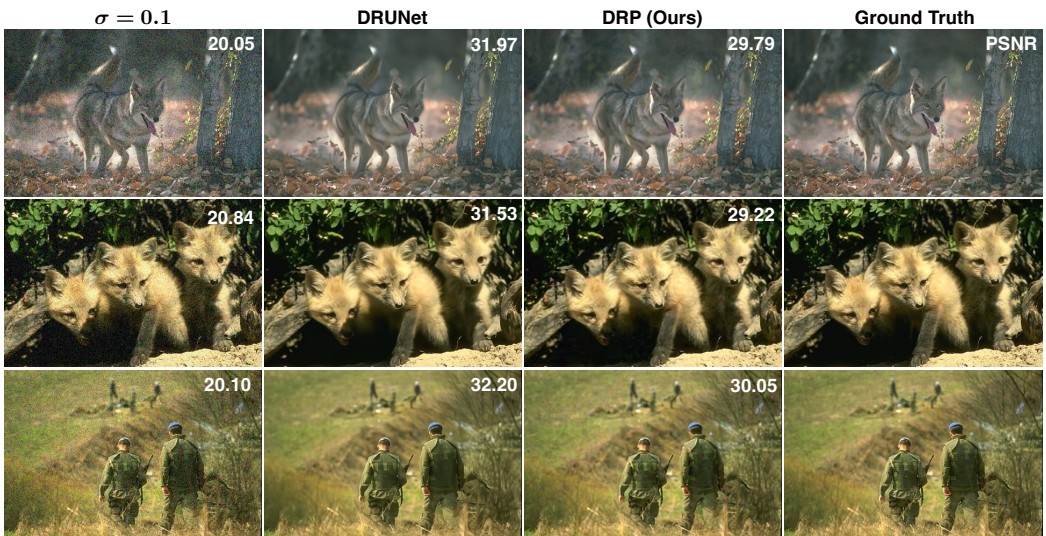

Figure 6: Illustration of denoising results of DRP on CBSD68 dataset with noise level $\sigma = 0.1$. Each image is labeled by its PSNR (dB) with respect to the original image.

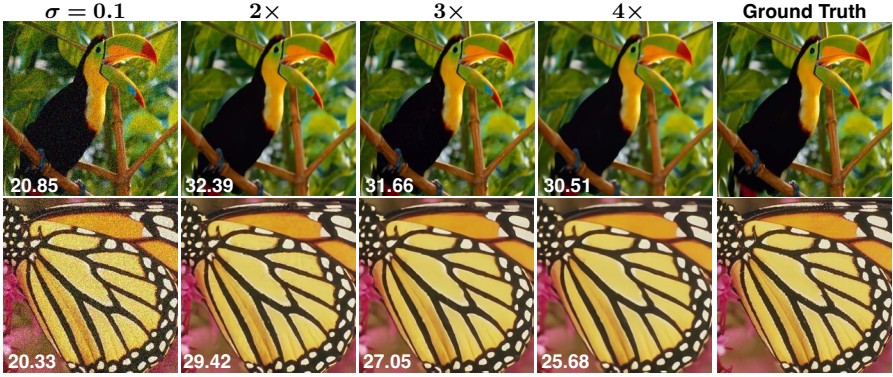

Figure 7: Illustration of denoising results of DRP on Set5 dataset with three SR level prior ($2\times$, $3\times$ and $4\times$). Each image is labeled by its PSNR (dB) with respect to the original image.

## B.2 COMPARISON WITH GS-PNP

In this subsection, we will compare DRP with the recent gradient-step denoiser PnP method (GS-PnP) (Hurault et al., 2022a). These comparisons were not included in the main paper due to space, but are provided here for completeness. GS-PnP provides comparable performance on image deblurring and single image super resolution as DPIR (Zhang et al., 2019), but comes with theoretical convergence guarantees.

Table 3 shows that DRP outperforms both DPIR and GS-PnP on image deblurring in most settings in terms of PSNR. Similarly, Table 4 shows that DRP can achieve better SISR performance in terms of PSNR compared to both methods. Figure 8 provides additional visual results on SISR showing that DRP can recover intricate details and sharpen features.

| Kernel | Datasets | GS-PnP | DPIR | DRP |
|---|---|---|---|---|
| | Set3c | 29.53 | 29.78 | **30.69** |
| | CBSD68 | 28.86 | 28.70 | **29.10** |
| | Set3c | 27.52 | 27.32 | **27.89** |
| | CBSD68 | 27.44 | **27.52** | 27.46 |

Table 3: PSNR performance of GS-PnP, DPIR, and DRP for image deblurring on Set3c and CBSD68 datasets on two blur kernels. The **best** and second best results are highlighted.

| SR | Kernels | Datasets | GS-PnP | DPIR | DRP |
|---|---|---|---|---|---|
| 2× | | Set3c | 28.23 | 28.18 | **29.26** |
| | | CBSD68 | 28.03 | 27.97 | **28.12** |
| | | Set3c | 26.19 | 26.80 | **27.41** |
| | | CBSD68 | 26.79 | 26.98 | **26.98** |
| 3× | | Set3c | 26.20 | 26.64 | **27.77** |
| | | CBSD68 | 26.77 | 26.80 | **27.18** |
| | | Set3c | 25.18 | 25.84 | **26.84** |
| | | CBSD68 | 26.30 | 26.39 | **26.60** |

Table 4: PSNR performance of GS-PnP, DPIR, and DRP for 2× and 3× SISR on the Set3c and CBSD68 datasets, using two blur kernels. The **best** and second best results are highlighted.

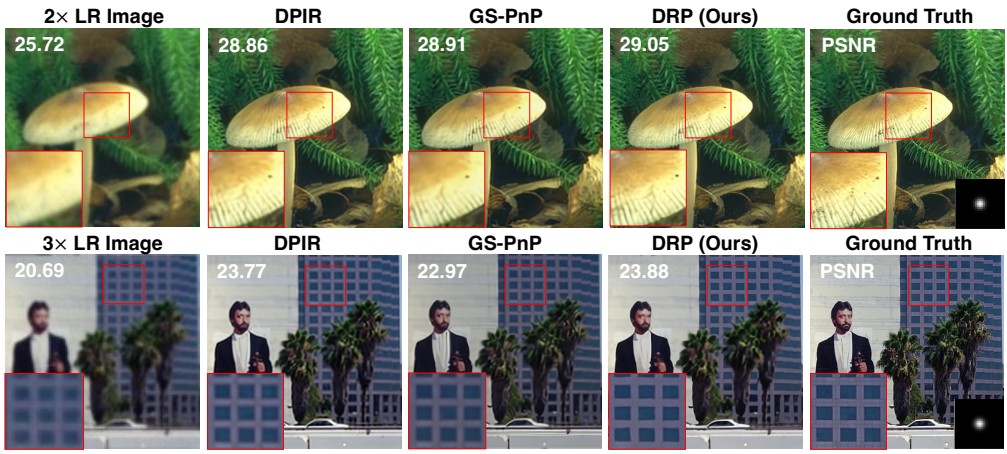

Figure 8: Illustration of SISR results of DRP compared with two SOTA denoiser based PnP method DPIR (Zhang et al., 2019) and GS-PnP (Hurault et al., 2022a). The top row displays the performance for 2× SISR task, while the bottom row showcases results for 3× SISR task. Each image is labeled by its PSNR (dB) with respect to the original image, and the visual difference is highlighted by the boxes in the left bottom corner.

## B.3 COMPARISON WITH DIFFUSION POSTERIOR SAMPLING

There is a growing interest in using denoisers within diffusion models for solving inverse problems (Zhu et al., 2023; Wang et al., 2022; Chung et al., 2023). One of the most wiedely-adopted diffusion model in this context is the *diffusion posterior sampling (DPS)* method from (Chung et al., 2023), which integrates pre-trained denoisers and measurement models for posterior sampling. One may argue that DPS is related to PnP due to the use of image denoisers as priors. In this section, we present results comparing DRP with DPS for deblurring human faces. We used the public implementation of DPS on the GitHub page that uses the prior specifically trained on human face image dataset (Chung et al., 2023). DRP uses the same SwinIR model trained on general image datasets (see Section 5.1). DPS and DRP are related but very different classes of methods. While DPS seeks to use denoisers to generate perceptually realistic solutions to inverse problems, DRP enables the adaptation of pre-trained restoration models as priors for solving other inverse problems.

Table 5 presents PSNR results obtained by DPS and DRP for human face deblurring. While we omitted the visual results from the paper for the privacy reasons, we will be happy to provide them if requested by the reviewers. Overall, DPS achieves more perceptually realistic images, while DRP achieves higher PSNR and more closely matches the ground truth images. This is not surprising when considering the generative nature of DPS. A similar observation is available in the original DPS publication, which reported better PSNR and SSIM performance of PnP-ADMM relative to DPS on SISR and deblurring (see Supplement E in (Chung et al., 2023)).

| Kernel | DPS | DRP |
|:------:|:---:|:---:|
|  | 29.61 | **34.61** |
|  | 28.80 | **33.05** |

Table 5: PSNR performance of DPS and DRP for image deblurring on three sample images from FFHQ validation set (provided in the DPS GitHub project) with two blur kernels. The **best** results are highlighted.

### B.4 PERFORMANCE OF SWINIR TRAINED FOR BICUBIC SR ON A MISMATCHED SISR TASK

In this section, we make a noteworthy point: the SwinIR SR network we used as a prior in our DRP method is specifically trained for the bicubic SR task. Its direct application to the SISR task (which is SR under Gaussian blur kernels and additive white Gaussian noise) leads to sub-optimal performance. This implies that our DRP method has the capacity to use a mismatched restoration model as an implicit prior, effectively adapting it for other image restoration tasks.

Figure 9 present qualitative and quantitative PSNR results on the Set3c dataset. Note how the direct use of SwinIR trained for bicubic SR does poorly on the SISR task, while using it within our DRP method as a prior leads to the SOTA performance.

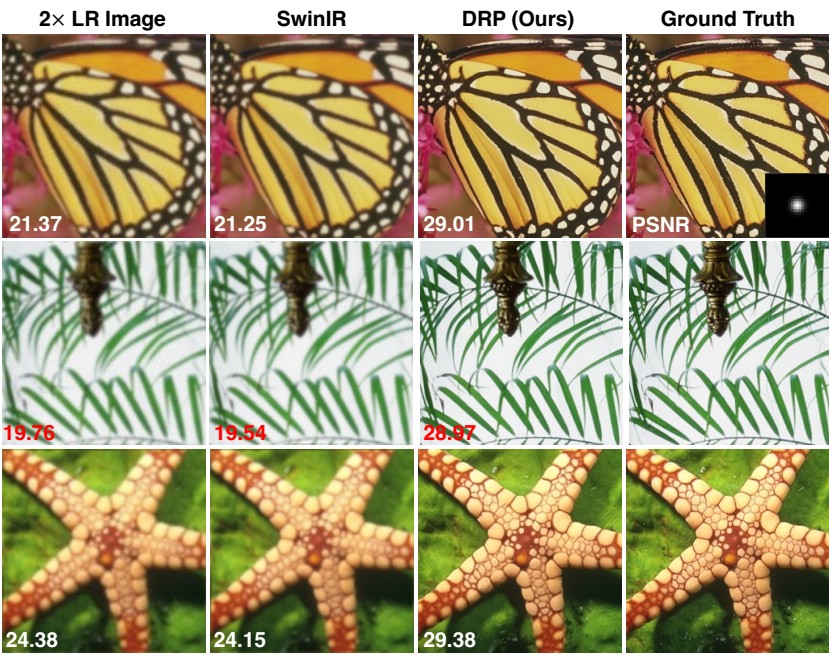

Figure 9: Illustration of $2\times$ SISR using DRP compared with the directly use of SwinIR trained for bicubic SR. Each image is labeled by its PSNR (dB) with respect to the ground truth.

## B.5 ADDITIONAL EVALUATION USING SSIM AND LPIPS QUALITY METRICS

In this section, we present results using two popular quality metrics—SSIM and LPIPS—for evaluating image restoration. We compare the proposed DRP method using SwinIR trained for bicubic SR as a prior against PnP-ADMM and DPIR. The results below complement those using PSRN in Table 1 and Table 2 in the main paper. As can be seen in Table 6 and Table 7, DRP always achieves better LPIPS and in most cases achieves better SSIM in all the debluring and SISR settings.

| Kernel | Datasets | PnP-ADMM | DPIR | DRP |
|---|---|---|---|---|
| | Set3c | 29.11/0.925/0.099 | 29.53/0.930/0.110 | **30.69/0.940/0.056** |
| | Set5 | 32.31/0.900/0.137 | 32.38/0.901/0.152 | **32.79/0.902/0.109** |
| | CBSD68 | 28.90/0.832/0.287 | 28.86/0.836/0.307 | **29.10/0.837/0.246** |
| | McMaster | 32.20/0.895/0.136 | 32.42/0.900/0.151 | **32.79/0.901/0.107** |
| | Set3c | 27.05/0.888/0.157 | 27.52/0.901/0.159 | **27.89/0.908/0.088** |
| | Set5 | 30.77/0.873/0.181 | 30.94/**0.880**/0.188 | **31.04**/0.875/**0.143** |
| | CBSD68 | 27.45/0.783/0.360 | **27.52/0.787**/0.376 | 27.46/0.773/**0.296** |
| | McMaster | 30.50/0.863/0.185 | 30.78/**0.871**/0.193 | **30.79**/0.865/**0.141** |

Table 6: PSNR↑/SSIM↑/LPIPS↓ of DRP and several SOTA PnP methods on image deblurring with the Gaussian blur kernels of standard deviations 1.6 and 2.0 on Set3c, Set5, CBSD68, and McMaster datasets. The **best** and second best results are highlighted. Note how DRP using image SR prior can outperform SOTA PnP methods that use denoisers as priors.

| SR | Kernel | Datasets | PnP-ADMM | DPIR | DRP |
|---|---|---|---|---|---|
| 2× | | Set3c | 27.88/0.903/0.131 | 28.18/0.908/0.137 | **29.26/0.915/0.079** |
| | | Set5 | 31.41/0.879/0.171 | 31.42/**0.883**/0.175 | **31.47**/0.876/**0.120** |
| | | CBSD68 | 28.00/0.796/0.338 | 27.97/**0.801**/0.356 | **28.12**/0.799/**0.285** |
| | | McMaster | 30.98/0.869/0.208 | 31.16/**0.875**/0.179 | **31.39**/0.873/**0.122** |
| | | Set3c | 25.86/0.861/0.187 | 26.80/0.884/0.171 | **27.41/0.891/0.107** |
| | | Set5 | 30.06/0.854/0.205 | 30.36/**0.862**/0.241 | **30.42**/0.859/**0.163** |
| | | CBSD68 | 26.88/0.749/0.393 | 26.98/**0.757**/0.401 | 26.98/0.745/**0.327** |
| | | McMaster | 29.53/0.836/0.219 | 29.87/**0.847**/0.215 | **30.03**/0.845/**0.167** |
| 3× | | Set3c | 25.85/0.865/0.173 | 26.64/0.883/0.169 | **27.77/0.898/0.090** |
| | | Set5 | 30.09/0.857/0.195 | 30.39/0.863/0.202 | **30.83/0.868/0.149** |
| | | CBSD68 | 26.78/0.751/0.378 | 26.80/0.755/0.394 | **27.18/0.763/0.299** |
| | | McMaster | 29.52/0.839/0.208 | 29.82/**0.848**/0.210 | **29.92**/0.841/**0.164** |
| | | Set3c | 24.87/0.834/0.216 | 25.84/0.863/0.195 | **26.84/0.879/0.108** |
| | | Set5 | 29.26/0.836/0.229 | 29.70/0.847/0.225 | **29.88/0.847/0.162** |
| | | CBSD68 | 26.28/0.722/0.420 | 26.39/0.730/0.422 | **26.60/0.735/0.324** |
| | | McMaster | 28.72/0.814/0.246 | 29.11/0.827/0.236 | **29.47/0.832/0.167** |

Table 7: PSNR↑/SSIM↑/LPIPS↓ comparison of DRP and several baselines for SISR on Set3c, Set5, CBSD68, and McMaster datasets. The **best** and second best results are highlighted. Note the excellent quantitative performance of DRP, which suggests the potential of using general restoration models as priors.

## B.6 COMPARISON ON SISR TASK WITH ADDITIONAL BASELINES

In this section, We will compare our proposed DRP method with some other baseline methods on $2\times$ SISR task. Specificity, we will compare it with DIP prior-based methods BSRDM (Yue et al., 2022) and pure learning-based method BSRNet (Zhang et al., 2021) on the same setting as we shown in Section 5.3. As observable in Table 8, DRP achieves better performance compared to both of them.

| SISR | Kernel | Datasets | BSRDM | BSRGAN | DRP |
|------|--------|----------|-------|--------|-----|
| $2\times$ | | Set3c | 23.51/0.825/0.139 | 22.69/0.818/0.102 | **29.26/0.915/0.079** |
| | | Set5 | 27.80/0.827/0.179 | 27.07/0.806/0.124 | **31.47/0.876/0.120** |
| | | CBSD68 | 25.66/0.718/0.378 | 25.51/0.723/**0.225** | 28.12/0.799/0.285 |
| | | McMaster | 27.43/0.804/0.226 | 26.78/0.785/0.132 | **31.39/0.873/0.122** |
| | | Set3c | 22.34/0.768/0.228 | 22.62/0.801/0.123 | **27.41/0.891/0.107** |
| | | Set5 | 26.68/0.791/0.246 | 26.88/0.789/**0.153** | 30.42/0.859/0.163 |
| | | CBSD68 | 24.86/0.670/0.451 | 25.24/0.695/**0.265** | 26.98/0.745/0.327 |
| | | McMaster | 26.47/0.768/0.294 | 26.69/0.770/**0.157** | 30.03/0.845/0.167 |

Table 8: PSNR↑/SSIM↑/LPIPS↓ comparison of DRP and several baselines for SISR on Set3c, Set5, CBSD68 and McMaster datasets. The **best** and second best results are highlighted. Note the excellent quantitative performance of DRP, which suggests the potential of using general restoration models as priors.

## B.7 COMPARISON WITH DPIR (SWINIR): DPIR EQUIPPED WITH SWINIR DENOISER

In this section,we will conduct an ablation study to shown the enhanced performance of DRP over denoiser-prior methods that do not originate from the SwinIR structure. To support this claim, we substitute the DRUNet denoiser utilized in DPIR with the SwinIR denoiser (Liang et al., 2021), called DPIR(SwinIR), aligning it with the same network structure employed in our SwinIR SR prior. As shown in table 9 and table 10, DRP can achieve better performance than both DPIR(SwinIR) and DPIR(DRUNet) in most of the debluring and SISR settings.

| Kernel | Datasets | DPIR (SwinIR) | DPIR (DRUNet) | DRP |
|---|---|---|---|---|
| | Set3c | 30.66/0.932/0.094 | 29.53/0.930/0.110 | **30.69/0.940/0.056** |
| | Set5 | 32.35/0.888/0.161 | 32.38/0.901/0.152 | **32.79/0.902/0.109** |
| | CBSD68 | 27.81/0.769/0.360 | 28.86/0.836/0.307 | **29.10/0.837/0.246** |
| | McMaster | 31.10/0.859/0.154 | 32.42/0.900/0.151 | **32.79/0.901/0.107** |
| | Set3c | 27.77/0.896/0.143 | 27.52/0.901/0.159 | **27.89/0.908/0.088** |
| | Set5 | 30.85/0.866/0.185 | 30.94/**0.880**/0.188 | **31.04**/0.875/**0.143** |
| | CBSD68 | 27.29/0.764/0.378 | **27.52/0.787**/0.376 | 27.46/0.773/**0.296** |
| | McMaster | 30.69/0.859/0.189 | 30.78/**0.871**/0.193 | **30.79**/0.865/**0.141** |

Table 9: PSNR (dB) of DRP and several SOTA methods for solving inverse problems using denoisers on image deblurring with the Gaussian blur kernels of standard deviation 1.6 and 2.0 on Set3c, Set5, CBSD68 and McMaster datasets. The **best** and second best results are highlighted. Note how DRP can outperform SOTA PnP methods that use denoisers as priors.

| SISR | Kernel | Datasets | DPIR(SwinIR) | DPIR(DRUNet) | DRP (SR) |
|---|---|---|---|---|---|
| 2× | | Set3c | 28.90/0.911/0.099 | 28.18/0.908/0.137 | **29.26/0.915/0.079** |
| | | Set5 | 31.38/0.872/0.122 | 31.42/**0.883**/0.175 | **31.47**/0.876/**0.120** |
| | | CBSD68 | 27.92/0.783/0.349 | 27.97/**0.801**/0.356 | **28.12**/0.799/**0.285** |
| | | McMaster | 31.34/0.870/0.127 | 31.16/**0.875**/0.179 | **31.39**/0.873/**0.122** |
| | | Set3c | 26.75/0.877/0.147 | 26.80/0.884/0.171 | **27.41/0.891/0.107** |
| | | Set5 | 30.23/0.852/0.240 | 30.36/**0.862**/0.241 | **30.42**/0.859/**0.163** |
| | | CBSD68 | 26.83/0.739/0.413 | 26.98/**0.757**/0.401 | 26.98/0.745/**0.327** |
| | | McMaster | 29.96/0.842/0.207 | 29.87/0.844/0.215 | **30.03/0.845/0.167** |
| 3× | | Set3c | 26.81/0.883/0.136 | 26.64/0.883/0.169 | **27.77/0.898/0.090** |
| | | Set5 | 30.35/0.855/0.159 | 30.39/0.863/0.202 | **30.83/0.868/0.149** |
| | | CBSD68 | 26.79/0.742/0.352 | 26.80/0.755/0.394 | **27.18/0.763/0.299** |
| | | McMaster | **29.94/0.845/0.193** | 29.82/**0.848**/0.210 | 29.92/0.841/**0.164** |
| | | Set3c | 25.66/0.853/0.174 | 25.84/0.863/0.195 | **26.84/0.879/0.108** |
| | | Set5 | 29.55/0.837/0.168 | 29.70/0.847/0.225 | **29.88**/0.847/**0.162** |
| | | CBSD68 | 26.29/0.715/0.431 | 26.39/0.730/0.422 | **26.60/0.735/0.324** |
| | | McMaster | 29.20/0.825/0.220 | 29.11/0.827/0.236 | **29.47/0.832/0.167** |

Table 10: PSNR↑/SSIM↑/LPIPS↓ comparison of DRP and several baselines for SISR on Set3c, Set5, CBSD68 and McMaster datasets. The **best** and second best results are highlighted. Note the excellent quantitative performance of DRP, which suggests the potential of using general restoration models as priors.

## B.8 EVALUATION OF DRP USING IMAGE DEBLURRING NETWORK AS A PRIOR

In this section, we show that DRP can be used with restoration priors and network architectures (beyond the SwinIR-based bicubic SR prior). To that end, we evaluate DRP under the deep unfolding network for image deblurring as a prior for solving Single Image Super Resolution (SISR) task.

### B.8.1 RESTORATION PRIOR BASED ON DEBLURRING DEEP UNFOLDING NETWORK

**Deep Unfolding Network Architecture.** We pre-trained a deblurring model using the USR-Net (Zhang et al., 2020) architecture based on deep unfolding network. Our training dataset consists of both the DIV2K (Agustsson & Timofte, 2017) and Flick2K (Lim et al., 2017) datasets, containing 3,450 color images in total. During training, we applied synthesized blur kernels to the input images, introducing Additive White Gaussian Noise (AWGN) characterized by $\sigma$ randomly chosen in [0, 5/255]. This process uses the same synthesized blur kernels as detailed in (Zhang et al., 2020).

The restoration network is trained only for the deblurring task, which limits its performance on the SISR task. The direct application of the deblur prior on SISR is presented under the name "Deblur" in Table 11. Note how the direc use of the deblurring network on SISR without DRP does not work.

### B.8.2 SINGLE IMAGE SUPER RESOLUTION USING DEBLURRING PRIOR

The measurement operator in SISR can be written as $\boldsymbol{A} = \boldsymbol{SK}$, where $\boldsymbol{K}$ is convolution with the blur kernel $\boldsymbol{k}$ and $\boldsymbol{S}$ performs standard $d$-fold down-sampling with $d^2 = n/m$. The scaled proximal operator $\mathsf{sprox}_{\lambda g}$ in (5) with data-fidelity term $g(\boldsymbol{x}) = \frac{1}{2}\|\boldsymbol{y} - \boldsymbol{SKx}\|_2^2$ can be write as:

$$\mathsf{sprox}_{\gamma g}(\boldsymbol{z}) = (\boldsymbol{K}^\mathsf{T}\boldsymbol{S}^\mathsf{T}\boldsymbol{SK} + \gamma\mathbf{H}^\mathsf{T}\mathbf{H})^{-1}[\boldsymbol{K}^\mathsf{T}\boldsymbol{S}^\mathsf{T}\boldsymbol{y} + \gamma\mathbf{H}^\mathsf{T}\mathbf{H}\boldsymbol{z}], \qquad (24)$$

where $\mathbf{H}$ is the convolution operator with blur kernel $\boldsymbol{k}$. Similarly to deblurring in Section 5.3, we use CG to efficiently compute (11). We adjust the hyper-parameter $\gamma$, $\tau$ and then use these parameters on the remaining datasets.

We refer to DRP using the Deblurring Prior as DRP (Deblur) and DRP using the Super-Resolution Prior as DRP (SR). Table 11 provides a comprehensive quantitative evaluation of the reconstruction performance for the $2\times$ Super-Resolution (SISR) task, using two distinct blur kernels. The results show that DRP (Deblur) outperforms DPIR and achieves a comparable performance as DRP (SR) across four datasets.

| | Kernel | Datasets | Deblur | DPIR | DRP (SR) | DRP (Deblur) |
|---|---|---|---|---|---|---|
| $2\times$ |  | Set3c | 13.63/0.300/0.640 | 28.18/0.908/0.137 | **29.26**/0.915/**0.079** | 29.13/**0.921**/0.082 |
| | | Set5 | 14.10/0.172/0.814 | 31.42/0.883/0.175 | 31.47/0.876/**0.120** | **31.81**/**0.883**/0.127 |
| | | CBSD68 | 13.69/0.152/0.849 | 27.97/0.801/0.356 | 28.12/0.799/0.285 | **28.27**/**0.704**/**0.283** |
| | | McMaster | 14.31/0.158/0.844 | 31.16/0.875/0.179 | 31.39/0.873/**0.122** | **31.52**/**0.881**/0.129 |
| |  | Set3c | 7.85/0.090/0.956 | 26.80/0.884/0.171 | 27.41/0.891/0.107 | **27.52**/**0.893**/**0.101** |
| | | Set5 | 8.77/0.052/1.00 | 30.36/0.862/0.241 | 30.42/0.859/**0.163** | **30.73**/**0.864**/0.175 |
| | | CBSD68 | 8.58/0.040/1.00 | 26.98/**0.757**/0.401 | 26.98/0.745/**0.327** | **27.19**/0.755/0.338 |
| | | McMaster | 9.26/0.048/1.00 | 29.87/0.844/0.215 | 30.03/0.845/**0.167** | **30.16**/**0.849**/0.179 |

Table 11: PSNR↑/SSIM↑/LPIPS↓ comparison of two DRP variants and several baselines for $2\times$ SISR on Set3c, Set5, CBSD68, and McMaster datasets. The **best** and second best results are highlighted. Note how the deblurring model doesn't generalize well on SISR without DRP; while its integration as a prior into DRP yields SOTA performance.

### B.9 Evaluation of Different Random Seeds

In this section, we present results illustrating the influence of different random seeds on the performance. We use three different random seeds for the image deblurring task on the Set3c dataset. For each seed, DRP is compared with the best baseline method DPIR on the image deblurring task.

| Seed ID | Kernel | DPIR | DRP |
|---------|--------|------|-----|
| ID = 0 | | 29.53/0.930/0.110 | **30.69/0.940/0.056** |
| ID = 10 | | 29.50/0.930/0.113 | **30.65/0.940/0.056** |
| ID = 100 | | 29.53/0.930/0.112 | **30.67/0.941/0.058** |
| ID = 0 | | 27.52/0.901/0.159 | **27.89/0.908/0.088** |
| ID = 10 | | 27.51/0.901/0.156 | **27.85/0.907/0.085** |
| ID = 100 | | 27.52/0.900/0.158 | **27.88/0.909/0.086** |

Table 12: PSNR↑/SSIM↑/LPIPS↓ performance of DPIR and DRP for image deblurring on the Set3c dataset with two blur kernels and three different random seeds. The **best** results are highlighted.

### B.10 Numerical Evaluation using Non-MMSE Priors

In this section, we show that although our theory relies on the assumption that the restoration prior used for inference performs MMSE estimation, in practice, DRP work even for priors trained with other loss functions, such as the $\ell_1$-norm or SSIM.

We retrained the SwinIR model for bicubic SR using the same setting as in Section 5.1, but with the $\ell_1$-norm loss instead of $\ell_2$. Table 13 shows that there is not much difference between the two. This suggests that DRP can be stable in practice even without the MMSE assumption.

| Kernel | DPIR | DRP ($SR_{\ell_1}$) | DRP ($SR_{\ell_2}$) |
|--------|------|---------------------|---------------------|
| | 29.53/0.930/0.110 | 30.67/**0.941/0.052** | **30.69**/0.940/0.056 |
| | 27.52/0.901/0.159 | 27.85/**0.909/0.086** | **27.89**/0.908/0.088 |

Table 13: PSNR↑/SSIM↑/LPIPS↓ performance of DRP($SR_{L1}$), DRP($SR_{L2}$) and DPIR for image deblurring on the Set3c dataset using two blur kernels. The **best** and second best results are highlighted.

