# OpenReview forum: "A Restoration Network as an Implicit Prior"
_ICLR.cc/2024/Conference — ICLR 2024 poster_

### Official Review · Reviewer_RzPk · 2023-10-26

**Soundness:** 2 fair
**Presentation:** 2 fair
**Contribution:** 2 fair
**Rating:** 5
**Confidence:** 4

**Summary:**

This paper aims at generalizing the idea of the implicit image prior parameterized by a pre-learned deep denoiser, for the plug-and-lay type image restoration. Specifically, the authors proposed the so-called deep restoration prior (DRP), which can be learned by more general image restoration tasks, such as super-resolution and deblurring. To demonstrate the effectiveness of the proposed method, the authors have trained a DRP based on super-resolution with a SwinIR architecture, and applied it to image deblurrring and super-resolution tasks.

**Strengths:**

1. The idea is interesting and provides some new perspectives of the implicit image prior.

2. Theoretical analysis has been provided.

**Weaknesses:**

1. The motivation of this work is not convincing enough. As I can understand, the aim of DRP is to better or more generally characterize the prior of images. However, it is doubtful whether this generalization is necessary since the deep denoising prior can be flexible enough by virtue of advanced neural network architectures.

2. The application of the proposed DRP is more complicated. Specifically, the additional degradation operator H can make the image restoration optimization process more complex, as can be observed in Algorithm 1. Besides, the additional operator H raises another question, that is how to choose this operator. The authors experimented with super-resolution prior, which assumes H to be downscaling, but it is unknown what if other operator is adopted, and which operator is the best.

3. The experimental comparison seems unfair. In specific, the authors used SwinIR for their DRP, while using DRUNet for DPIR, though DRUNet was adopted in the original DPIR work. It is necessary to re-implement DPIR with SwinIR, or re-implement DRP with DRUNet, such that the influences of network structure can be controlled.

**Questions:**

My concerns are provided in the "weaknesses" part.

---

> ### Author Response · Authors · 2023-11-19
> **Response to reviewer RzPk**
>
> Thank you for your valuable feedback and positive comments on our work. Please see below for our point-by-point responses to your comments.
>
> >1: The motivation of this work is not convincing enough. As I can understand, the aim of DRP is to better or more generally characterize the prior of images. However, it is doubtful whether this generalization is necessary since the deep denoising prior can be flexible enough by virtue of advanced neural network architectures.
>
> Our work is the first to generalize the notion of the priors beyond denoising networks. Despite their popularity, nothing in the existing theory says that denoisers are better priors in general. While denoisers are a special case of our formulation (H = I), our work opens the door to other restoration networks that were not considered before (H ≠ I). In fact, our work provides concrete evidence that the SR networks can be better than denoising networks on some tasks.
>
> Our motivation goes beyond “advanced network architectures”. One can train the same architecture—such as SwinIR—for denoising or the SR task. However, denoiser and SR models will give different priors and different results, despite corresponding to the same architecture. Prompted by your comment, Section B.7 shows the performance of DPIR using SwinIR trained as a denoiser. Our DRP uses the same SwinIR architecture, but trained to perform SR. Note how DRP does better than DPIR, despite both using priors with **identical network architectures**.
>
> >2: The application of the proposed DRP is more complicated. Specifically, the additional degradation operator H can make the image restoration optimization process more complex, as can be observed in Algorithm 1. Besides, the additional operator H raises another question, that is how to choose this operator. The authors experimented with super-resolution prior, which assumes H to be downscaling, but it is unknown what if other operator is adopted, and which operator is the best.
>
> The computational overhead of DRP is small compared to the traditional denoiser-based methods (such as DPIR or PnP-ADMM). Evaluating sprox in our DRP method has roughly the same complexity as evaluating prox in DPIR.
>
> One controbution of our work is that it opens the question on the choice of H. As you correctly noted, this question was ***not even considered*** before, since the research in the area was focused on using denoisers as priors. Our work challenges this paradigm and presents hard evidence that one can do better by considering other restoration networks than image denoisers.
>
> Prompted by your comment, we ran new experiments by including another restoration prior corresponding to a deep unfolding network for solving the image deblurring task. The results are presented in Section B.8 of the supplement. Table 11 shows that the new deblurring prior cannot directly handle the SISR task; yet, it is an excellent prior for SISR within our DRP method.
>
> >3: The experimental comparison seems unfair. In specific, the authors used SwinIR for their DRP, while using DRUNet for DPIR, though DRUNet was adopted in the original DPIR work. It is necessary to re-implement DPIR with SwinIR, or re-implement DRP with DRUNet, such that the influences of network structure can be controlled.
>
> Prompted by your comment, we re-implemented DPIR using SwinIR trained as a denoiser. These results are presented in Section B.7. DRP uses the same SwinIR architecture as the new DPIR (SwinIR), but the SwinIR in DRP is trained to perform SR. Note how DRP still does better than DPIR, despite both using priors with identical network architectures.

---

> > ### Comment · Reviewer_RzPk · 2023-11-19
> >
> > Thanks for your reply!
> >
> > The new results provided in B.7 seem to suggest that DRUNet is better for SwinIR when applied to PnP prior, at least for DPIR. Have tried to implement your DRP with DRUNet?

---

> ### Author Response · Authors · 2023-11-19
> **On the use DRUNet for DRP**
>
> Thank you for reading our response and providing additional thoughts.
>
> > The new results provided in B.7 seem to suggest that DRUNet is better for SwinIR when applied to PnP prior, at least for DPIR. Have tried to implement your DRP with DRUNet?
>
> Do you mean **(a)** to use a DRUNet denoiser within DRP or **(b)** train DRUNet for SR and use it as a SR prior within DRP?
>
> **(a):** It would not make conceptual sense to use a denoiser within DRP, as DRP is designed for other restoration networks. If we set H = I within DRP and use DRUNet, we simply turn DRP into a PnP method similar to DPIR or GS-PnP.
>
> **(b):** While we could train DRUNet for SR and use it as a prior within DRP, it is not clear what this experiment would achieve. Whether DRUNet architecture is better than SwinIR architecture is tangential to our main point. Trying different architectures would not change the key point of our paper, since it is not about specific architectures; it rather states that restoration models—beyond image denoisers—can be used as image priors. Our theory specifies the corresponding regularizer in eq. (9) and our simulations provide evidence that a SR prior can outperform denoisers.

---

> > ### Comment · Reviewer_RzPk · 2023-11-20
> >
> > I mean Case (b), and I believe this experiment is meaningful, since it can in some sense validate whether the proposed method is REALLY "not about specific architectures".

---

> ### Author Response · Authors · 2023-11-20
> **Thank you for additional thoughts**
>
> Thank you for your continuous interest.
>
> > I mean Case (b), and I believe this experiment is meaningful, since it can in some sense validate whether the proposed method is REALLY "not about specific architectures".
>
> **Note that there is no point in using a worse SR architecture as a prior. The idea of DRP is to be able to use SOTA restoration architectures as priors for solving inverse problems.**
>
> For the sake of argument consider both possible outcomes:
>
> **Oucome 1:** DRP with SwinIR > DRP with DRUNet [more likely]: Conclusion will be that the SwinIR architecture gives a better SR prior for DRP. DRP with SOTA SR prior still outperforms SOTA PnP baselines based on Gaussian denoisers as priors.
>
> **Outcome 2:** DRP with DRUNet > DRP with SwinIR [less likely]: Conclusion will be that the DRUNet architecture gives a better SR prior for DRP. DRP with SOTA SR prior still outperforms SOTA PnP baselines based on Gaussian denoisers as priors.
>
> Note how our formulation and theory are independent of a specific architecture, giving our DRP method an ability to "plug-and-play" different architectures. If tomorrow someone invents an even better architecture for SR than SwinIR or DRUNet, we can directly use it within DRP for solving inverse problems.
>
> If the Reviewer insists, we will be delighted to run this experiment for them (it will take some time for us to run this and report results).

---

> ### Comment · Reviewer_RzPk · 2023-11-21
>
> So we may conclude that the effectiveness of the implicit DO depends on the network architecture?  Then we could expect to have a better denoising prior other than SwinIR and DRUNet, such that the performance of DPIR can be better. Considering this point, the motivation for this work is still not very convincing.

---

> ### Author Response · Authors · 2023-11-21
> **DRP is not restricted to Gaussian denoisers, hence it is more general**
>
> > So we may conclude that the effectiveness of the implicit DO depends on the network architecture? Then we could expect to have a better denoising prior other than SwinIR and DRUNet, such that the performance of DPIR can be better. Considering this point, the motivation for this work is still not very convincing.
>
> We hope the Reviewer considers the following convincing motivation:
>
> **DPIR is limited to Gaussian denoisers as priors. Our DRP is *not* limited to Gaussian denoisers, making DRP an inherently more general method.**
>
> There is no reason to conceptually limit ourselves to Gaussian denoiser priors. We show cases where DRP is clearly better than  DPIR for two widely-used architectures – SwinIR and DRUNet (see Section B.7 in the supplement). We could adopt other network architectures for both DRP and DPIR, and we are confident DRP can always do great, *since it is more general*.

---

> > ### Author Response · Authors · 2023-11-21
> > **Completed experiments using the DRUNet SR network**
> >
> > > The new results provided in B.7 seem to suggest that DRUNet is better for SwinIR when applied to PnP prior, at least for DPIR. Have tried to implement your DRP with DRUNet?
> >
> > Dear Reviewer, we have finally completed the requested experiment using the DRUNet architecture trained for SR as a SR prior within DRP. The new results are presented in Tables 11 and 12 of Section B.8. The new DRP (DRUNet SR) attains comparable performance to DRP (SwinIR SR), consistently outperforming DPIR (DRUNet denoiser).
> >
> > The new results confirm the effectiveness of DRP and provide experimental validation for its robustness across different SR network architectures (both SwinIR and DRUNet).
> >
> > **We hope that with this new experiment and the fact that we have addressed all your comments, you will reconsider your evaluation of our work and raise the score.**

---

> > > ### Comment · Reviewer_RzPk · 2023-11-22
> > >
> > > Thanks for your reply. Your responses have addressed several concerns of mine, and I can realize the contribution of this work and decide to raise my rating. However, I still do not believe the contribution is significant enough for a conference such as ICLR.

---

> > > > ### Author Response · Authors · 2023-11-22
> > > > **Response to Reviewer RzPk**
> > > >
> > > > Thank you for reading our comments and raising the score. While we disagree with you on the significance of our work—as we think that it is significant enough for ICLR—we appreciate the feedback that you have provided.

---

### Official Review · Reviewer_3U6y · 2023-10-29

**Soundness:** 3 good
**Presentation:** 3 good
**Contribution:** 3 good
**Rating:** 6
**Confidence:** 4

**Summary:**

This work proposes the DRP method which utilizes a pre-trained restoration network as a prior to solve the inverse problem. Under several mild assumptions, it proves the convergence of DRP. In the experiments, two popular tasks (e.g., debluring and super-resolution) are considered to validate the effectiveness of the proposed method.

**Strengths:**

1. The paper is well written, and easy to follow.
2. It provides sound theory proof on the convergence of DRP.

**Weaknesses:**

1. As for evaluation metric, more commonly-used metrics should be employed to have a comprehensive comparison, such as SSIM, LPIPS.
2. More comparative methods should be considered, including the DIP prior-based methods DIPFKP (CVPR 2021) and BSRDM (CVPR 2022) and some directly trained based method, such as BSRNet (ICCV 2021), RealESRNet (ICCV 2021 workshop).
3. This is a non-blind method. The experiments only verify its simple case with known degradation. I wonder that is it able to handle the real-world case?
4.  As shown in the appendix, its performance is obviously inferior to DRUNet. Additionally, it relies on the SwinIR as a prior. It is necessary to conduct a comparison with SwinIR.

**Questions:**

1. According to my understanding, the motivation and idea of this work is very similar to IRCNN. The main difference to IRCNN is that it uses a more powerful restoration network SwinIR instead of DnCNN. The sprox operator in Eq. (5) corresponds to the sub-optimization problem Eq. (6a) of the paper of IRCNN. The core step 3 in Algorithm 1 corresponds to Eq. (6b) of IRCNN.
2. Following the last question, step 3 introduce the restoration prior in Algorithm 1. I'm interested in that how to induce the updated procedure of step 3. In other word, a more intuitive explanation on step 3 should be provided.

---

> ### Author Response · Authors · 2023-11-19
> **Response to reviewer 3U6y**
>
> Thank you for your valuable feedback and positive comments on our work. Please see below for our point-by-point responses to your comments.
>
> >1: As for evaluation metric, more commonly-used metrics should be employed to have a comprehensive comparison, such as SSIM, LPIPS.
>
> This is an excellent suggestion. Prompted by your comment, we report SSIM and LPIPS as additional evaluation metrics in Section B.5 of the supplement. Note the excellent performance of DRP for all metrics – PSNR, SSIM, and LPIPS – on all tasks.
>
> >2: More comparative methods should be considered, including the DIP prior-based methods DIPFKP (CVPR 2021) and BSRDM (CVPR 2022) and some directly trained based method, such as BSRNet (ICCV 2021), RealESRNet (ICCV 2021 workshop).
>
> Prompted by your comment, the revised manuscript considers two additional methods in Section B.6, including the DIP-based BSRDM and the directly-trained method BSRGAN. Note how in Table 8, DRP achieves better performance compared to both of these methods.
>
> >3: This is a non-blind method. The experiments only verify its simple case with known degradation. I wonder that is it able to handle the real-world case?
>
> As you correctly observed, DRP is designed to address non-blind inverse problems. Such problems are common in many applications, including microscopy and medical imaging. Extension of DRP to blind inverse problems would be an exciting research direction for the future. For example, one can imagine an extension that allows the update of the degradation operator during the inference of the DRP algorithm. We state in the “Limitations and Future Work” section of the revised manuscript our interest in the extension to blind inverse problems.
>
> >4: As shown in the appendix, its performance is obviously inferior to DRUNet. Additionally, it relies on the SwinIR as a prior. It is necessary to conduct a comparison with SwinIR.
>
> Section B.1 of the supplement does not imply that our method is in general inferior to DRUNet. It shows that SwinIR trained for bicubic SR can be used as a prior for denoising, but **as expected** it is worse than DRUNet specifically trained for denoising. This is better clarified in the revised Section B.1 .
>
> In Section B4, we presented the SISR performance with and without DRP. Notably, SwinIR itself struggles with the SISR task in the absence of DRP. However, with the integration of DRP, we achieve state-of-the-art performance.
>
> What’s more, In Section B.7, we reimplement DPIR using SwinIR denoiser and add it as a new ablation study for all experiment settings. Table 9 and Table 10 in Section B.7 of the supplement show that DRP with SwinIR SR prior can consistently outperforms DPIR with SwinIR denoiser prior.
>
> >5:  According to my understanding, the motivation and idea of this work is very similar to IRCNN. The main difference to IRCNN is that it uses a more powerful restoration network SwinIR instead of DnCNN. The sprox operator in Eq. (5) corresponds to the sub-optimization problem Eq. (6a) of the paper of IRCNN. The core step 3 in Algorithm 1 corresponds to Eq. (6b) of IRCNN.
>
> There has been a lot of interest in using priors given as networks trained to perform image denoising. IRCNN, PnP-ADMM, RED, GS-PnP, and DPIR are all similar in the sense that they are based on this same principle.
>
> Our work is the first to generalize this idea to priors given as networks trained to perform other restoration tasks. Our method is similar to IRCNN only when H=I; however, unlike IRCNN, our method allows for H ≠ I, making it compatible with many more priors.
>
> The main difference is not that we use a “more powerful network” SwinIR, but that we use a network trained for a different task. Prompted by your comment, Section B.7 shows the performance of DPIR—a more recent variant of IRCNN—using SwinIR trained as a denoiser. Our DRP method uses the same SwinIR architecture, but trained to perform SR. Note how DRP still does better than DPIR, despite both using priors with ***identical network architectures***.
>
> >6: Following the last question, step 3 introduce the restoration prior in Algorithm 1. I'm interested in that how to induce the updated procedure of step 3. In other word, a more intuitive explanation on step 3 should be provided.
>
> We added a more detailed explanation in Section 5.2 of the paper. As a  bicubic SR model is adopted as prior in the setting of Section 5.2, the R(Hx) in Step 3 of Algorithm 1 performs a bicubic downsampling of the intermediate image and inputs it into the bicubic SR SwinIR.

---

> > ### Comment · Reviewer_3U6y · 2023-11-20
> > **Response to authors**
> >
> > I still have some concerns on the following questions:
> >
> >  Q4: As shown in the appendix, its performance is obviously inferior to DRUNet. Additionally, it relies on the SwinIR as a prior. It is necessary to conduct a comparison with SwinIR.
> >
> > The comparsion with SwinIR on Fig.9 is not fair due to the degradation mismatch. As for as I know, there is a public SwinIR model trained on the general degradation for  SISR. The author should use such a version of SwinIR as a prior to validate the effectiveness of the proposed framework.
> >
> > Q5. Similiarity with DPIR or IRCNN.
> >
> > IRCNN is also adopted well for the case of H=/ I, such as the task of SISR and deblurring. I still think this is an incremental work on IRCNN otherwise you can distinguish from it. The core step 3 in Algorithm 1 corresponds to Eq. (6b) of IRCNN.

---

> ### Author Response · Authors · 2023-11-20
> **Thank you for reading our response**
>
> Thank you for reading our response and providing additional thoughts.
>
> > The comparsion with SwinIR on Fig. 9 is not fair due to the degradation mismatch. As for as I know, there is a public SwinIR model trained on the general degradation for SISR. The author should use such a version of SwinIR as a prior to validate the effectiveness of the proposed framework.
>
> Fig. 9 is not really a "comparison with SwinIR" as DRP is using the same SwinIR as a prior. Fig. 9 illustrates how the **same** SwinIR model can work with and without DRP when applied to a *mismatched* task. A similar idea is illustrated in Section B.9 using another network that is not SwinIR.
>
> We would like the Reviewer to consider our main results in Section 5 that show how SwinIR can be an excellent prior competitive with Gaussian denoisers traditionally used in DPIR, IRCNN, or other PnP methods.
>
> **If the Reviewer is concerned that Fig. 9 would confuse readers, we can certainly remove it from the paper.**
>
> > IRCNN is also adopted well for the case of H $\neq$ I, such as the task of SISR and deblurring. I still think this is an incremental work on IRCNN otherwise you can distinguish from it. The core step 3 in Algorithm 1 corresponds to Eq. (6b) of IRCNN.
>
> We see the source for the Reviewer's confusion. Let us clarify the differences step-by-step:
>
> **(a)** First note that the notation H in the IRCNN paper corresponds to A in our paper (see eq. (1) in our paper) – and **not** to H in our paper. This is the main source for the Reviewer's confusion. This operator A denotes the actual inverse problem we are trying to solve, such as SISR and deblurring.
>
> **(b)** IRCNN only uses Gaussian denoisers to solve an inverse problem (this is clearly stated in Section 3 of the IRCNN paper). Our DRP uses more general restoration operators than denoisers to solve the inverse problem. Our paper uses H to denote the prior restoration operator that we use to solve the inverse problem given by A.
>
> **(c)** If we adopt the notations in our paper, IRCNN is only similar to DRP when H = I. This is obvious if you compare eq. (7) in the IRCNN paper with eq. (10) in our paper. In eq. (10), we have two operators – the measurement operator A = K and the prior degradation operator H—while eq. (7) in the IRCNN paper only has the measurement operator.
>
> **(d)** Note that eq. (6b) of IRCNN is the traditional "proximal operator" that is replaced by a Gaussian denoiser in eq. (9) of IRCNN. This step is done in *all* PnP methods, including DPIR and IRCNN. Our work is novel because we do *not* use Gaussian denoisers as priors.
>
> Our work is the first principled generalization of PnP methods – including IRCNN – beyond Gaussian denoisers. We provide a novel and more general formulation, establish a novel theory in Theorems 1 and 2, and give new evidence that one can do better by using priors beyond Guassian denoisers.
>
> **We hope that now it is clear to the Reviewer that our work is certainly *not* an "incremental work on IRCNN".**

---

> > ### Comment · Reviewer_3U6y · 2023-11-20
> > **Response to authors**
> >
> > Thanks for your reply.
> >
> > I understand the settings of mismatch degradation in Fig. 9. However, I think it is necessary to verify the margin performance gain brought by the proposed method under the same degradation settings, e.g., to solve the general SISR problem based on a pretrained SwinIR model under the general degradations.

---

> ### Author Response · Authors · 2023-11-20
> **Thank you for your continuous interest**
>
> Thank you for your continuous interest.
>
> > I understand the settings of mismatch degradation in Fig. 9. However, I think it is necessary to verify the margin performance gain brought by the proposed method under the same degradation settings, e.g., to solve the general SISR problem based on a pretrained SwinIR model under the general degradations.
>
> Since you are familiar with IRCNN, let me make an analogy. IRCNN uses a Gaussian denoiser as a prior (see the CNN architecture in Figure 1 of the IRCNN paper) to solve *other* restoration tasks. However, the IRCNN paper *never* compares that CNN denoiser against IRCNN on Gaussian denoising to show in the words of the Reviewer the "gain brought by the proposed method [IRCNN] under the same degradation settings [Gaussian denoising]".
>
> We have a conceptually similar setting – DRP uses SwinIR as a SR restoration prior. We show with extensive experiments that SwinIR can be used to solve *other* inverse problems.
>
> It is clear that using IRCNN on Gaussian denoising (matched task) will perform about the same as using the Gaussian denoiser directly. This is because (a) IRCNN will *always* perform at least as good as its denoiser prior and (b) there is no new information brought by IRCNN in the matched case (when H = I in the notation of the IRCNN paper).
>
> The same logic applies to our work. Using DRP for matched restoration problem will perform about the same as using SwinIR because (a) DRP will *always* perform at least as good as its SR prior and (b) there is no new information brought by DRP in the matched case (when A = H in our paper).
>
> **Please see below for the requested comparison with a SwinIR model pretrained under the general degradations.**

---

> ### Author Response · Authors · 2023-11-20
> **Updated Figure 9 with a comparison with a SwinIR model under the general degradations**
>
> > I understand the settings of mismatch degradation in Fig. 9. However, I think it is necessary to verify the margin performance gain brought by the proposed method under the same degradation settings, e.g., to solve the general SISR problem based on a pretrained SwinIR model under the general degradations.
>
> Prompted by your request, we updated Figure 9 to include an additional comparison with an official SwinIR model pre-trained using general degradations (from https://github.com/JingyunLiang/SwinIR). As expected, SwinIR (RealSR) is much worse than DRP. We hope that this addresses the Reviewer's comment.

---

> > ### Comment · Reviewer_3U6y · 2023-11-21
> > **Response to Authors**
> >
> > My concern was addressed. I increase my rating.

---

> ### Author Response · Authors · 2023-11-21
> **Response to the Reviewer**
>
> We appreciate your positive engagement throughout the discussion period. Please let us know if there is anything else we can answer or clarify that would lead to further improve your evaluation of our paper.

---

### Official Review · Reviewer_U9fM · 2023-10-31

**Soundness:** 4 excellent
**Presentation:** 4 excellent
**Contribution:** 3 good
**Rating:** 8
**Confidence:** 5

**Summary:**

In plug-and-play-type algorithms, a denoiser (such as a CNN trained to denoise images) is plugged in the place of the proximal operator which is part of the formally derived algorithm. This paper proposes a way of extending this approach in a way that allows the use not only denoisers but also other restoration networks (e.g., a trained super-resolution CNN). The extension is simple, although non-trivial, and the paper provides theoretical convergence guarantees and experimental validation.

**Strengths:**

As I mentioned above, the extension proposed in this paper is simple, although non-trivial, and the paper provides theoretical convergence guarantees and experimental validation. It is a solid paper and the authors show a clear and solid knowledge of the field and of the state of the art, which is comprehensively reviewed. In summary, although certainly not a breakthrough or a very exciting new method, it is a good quality piece of work.

**Weaknesses:**

Although this is a good quality paper, there are a few aspects that could be improved, some important, others less so.

First, and most importantly, the paper lacks some discussion/analysis of why restoration networks for problems other than denoising can lead to better results than denoising regularizers. Do they learn better image priors? This is somewhat surprising, in that these other networks are expected to have learned (in addition to a "prior") also specific aspects of the particular problem in which they were trained. This begs the question: what problems yield the best restoration networks? Is this choice related to the main problem to be solved?

The previous comment is related to the following observation: although the authors present the method as working for general restoration networks, they end up only using restoration networks trained for super-resolution. Arguably, SR is a very particular type of inverse problem, in a sense, the one that is closer to pure denoising than any other inverse problem. The authors should comment on this.

In Section 4, there is some confusion regarding necessary and sufficient conditions. The authors write "Our analysis will require several assumptions that act as sufficient conditions for our theoretical results." This sentence is self-contracting: if the assumptions are *required* they are *necessary* ("necessary" and "required" are synonyms). In fact, further down, the authors contradict that sentence, by writing "This mild assumption is necessary ...".

The first inequality in Theorem 2 is trivial and doesn't even need to be mentioned.

Minor style issue: it is not good style to write "...discussion in Kamilov et al. (2023) on ..." or "... in Chapter 3 in Beck (2017) ...". Kamilov and Beck are not papers or books, but people. A nicer way to write these sentences is "...discussion in the work of Kamilov et al. (2023) on ..." or "...discussion in the paper by Kamilov et al. (2023) on ..." and "... in Chapter 3 of the book by Beck (2017) ...".

A probability density function is degenerate, not only if it is supported on a subspace, but on any zero-measure set in the ambient space, for example, a d-manifold, with d < n.

Typo: "Our method is as a major extension ..." should be "Our method is a major extension ...".

**Questions:**

The form of Equation (9) suggests that using the regularizer trained to solve (4) is in some sense equivalent, or at least related, to analysis regularization/priors, where $p_s$ is a learned prior and ${\bf H}$ the analysis operator. See https://doi.org/10.1117/12.826663
What do the authors think of this connection?

How much is lost by solving (10) and (11) by CG and how do the authors know that 3 iterations is enough? Would it be worth (at least for (10)) to give some careful thought to the form of the inversions needed; maybe it is possible to exploit the fact that bot ${\bf H}$ and ${\bf K}$ can are convolutions to obtain closed-form inversions using FFTs.

**Details Of Ethics Concerns:**

Not applicable.

---

> ### Author Response · Authors · 2023-11-19
> **Response to reviewer U9fM (1/2)**
>
> Thank you for your valuable feedback and positive comments on our work. Please see below for our point-by-point responses to your comments.
>
> >1: First, and most importantly, the paper lacks some discussion/analysis of why restoration networks for problems other than denoising can lead to better results than denoising regularizers. Do they learn better image priors? This is somewhat surprising, in that these other networks are expected to have learned (in addition to a "prior") also specific aspects of the particular problem in which they were trained. This begs the question: what problems yield the best restoration networks? Is this choice related to the main problem to be solved?
>
> This is an excellent observation. As you correctly noted, by suggesting to consider restoration networks beyond image denoisers, our work opens an intriguing question of what is the **best** restoration network? This question was not considered before since the research in the area was primarily focused on using denoisers as priors. While we cannot yet answer the question of optimality, our paper provides concrete evidence that for some tasks other restoration networks are better priors than denoisers. We added a short discussion on this in the “Limitations and Future Work” section.
>
> >2: The previous comment is related to the following observation: although the authors present the method as working for general restoration networks, they end up only using restoration networks trained for super-resolution. Arguably, SR is a very particular type of inverse problem, in a sense, the one that is closer to pure denoising than any other inverse problem. The authors should comment on this.
>
> Indeed, it would be interesting to explore the performance of DRP for more priors, beyond the SR network. Prompted by your comment, we ran new experiments by including another prior corresponding to a deep unfolding network for solving image deblurring. The results are presented in Section B.9 of the supplement. As can be seen in Table 13, the deblurring network can serve as an excellent prior for regularizing SISR within our DRP method. We are certainly interested in exploring the performance of DRP under even more priors, but will leave this to future work due to limited rebuttal time.
>
> >3: In Section 4, there is some confusion regarding necessary and sufficient conditions. The authors write "Our analysis will require several assumptions that act as sufficient conditions for our theoretical results." This sentence is self-contracting: if the assumptions are required they are necessary ("necessary" and "required" are synonyms). In fact, further down, the authors contradict that sentence, by writing "This mild assumption is necessary ...".
>
> We have reworded our statements for clarity.
>
> >4: The first inequality in Theorem 2 is trivial and doesn't even need to be mentioned.
>
> We have removed the first inequality from Theorem 2.
>
> >5: Minor style issue: it is not good style to write "...discussion in Kamilov et al. (2023) on ..." or "... in Chapter 3 in Beck (2017) ...". Kamilov and Beck are not papers or books, but people. A nicer way to write these sentences is "...discussion in the work of Kamilov et al. (2023) on ..." or "...discussion in the paper by Kamilov et al. (2023) on ..." and "... in Chapter 3 of the book by Beck (2017) ...".
>
> Prompted by your comment, we updated our style to always use (Author, Year) rather than Author (Year). We also revised corresponding sentences to make sense.
>
> >6: A probability density function is degenerate, not only if it is supported on a subspace, but on any zero-measure set in the ambient space, for example, a d-manifold, with d < n.
>
> Indeed, we revised the statement to clarify that “subspace” is only one example.
>
> >7: Typo: "Our method is as a major extension ..." should be "Our method is a major extension ...".
>
> We have fixed the typo.
>
> >8: The form of Equation (9) suggests that using the regularizer trained to solve (4) is in some sense equivalent, or at least related, to analysis regularization/priors, where there is a learned prior and the analysis operator. See https://doi.org/10.1117/12.826663 What do the authors think of this connection?
>
> While we didn't initially see the connection to analysis/synthesis priors, your comment provides a new perspective to our work. One can indeed view Eq. (9) in our formulation as an analysis regularizer with a transform H. This might open new directions for designing data-driven priors. We mention this new connection in the revised paper and cite the paper you shared there.

---

> ### Author Response · Authors · 2023-11-19
> **Response to reviewer U9fM (2/2)**
>
> >9: How much is lost by solving (10) and (11) by CG and how do the authors know that 3 iterations is enough? Would it be worth (at least for (10)) to give some careful thought to the form of the inversions needed; maybe it is possible to exploit the fact that bot H and K can are convolutions to obtain closed-form inversions using FFTs.
>
> Indeed, one can design dedicated closed form implementations of the scaled proximal operator for some linear transforms. The reason we used CG was due to its generality, in the sense that the same implementation can be used for all linear transforms. We didn’t observe any improvements from a higher number of CG sub-iterations, which suggests that 3 is sufficient for our experiments. Note that it is common to use 2-5 iterations of CG within proximal algorithms by using outer iterations to initialize CG (see, for example, the discussion on page 88 of (Kamilov et al., 2023)).

---

> ### Author Response · Authors · 2023-11-21
> **Response to Reviewer U9fM**
>
> Dear Reviewer U9fM, we appreciate your positive evaluation of our work. As we near the deadline, please let us know if there is something we can do to further improve your evaluation.

---

### Official Review · Reviewer_v5mr · 2023-11-01

**Soundness:** 3 good
**Presentation:** 2 fair
**Contribution:** 3 good
**Rating:** 6
**Confidence:** 4

**Summary:**

The authors propose to improve a family of solutions to inverse problems called plug and play reconstruction methods by replacing their implicit priors. Instead of utilizing only denoisers, the proposed method suggests other reconstruction methods can also be utilized as priors. They provide a theoretical study with key assumptions to illustrate the convergence. They present a numerical study on the convergence for two applications considered, deblurring and super-resolution. Lastly, the authors present results on multiple datasets compared against related previous works, showing improvements in most cases, according to their evaluation method.

**Strengths:**

Originality: The main contribution of the work is extending the formulation of plug and play reconstruction methods to utilize implicit priors other than only denoisers.

Clarity: The theoretical analysis of their proposed idea is clearly presented.

Significance: While the performance numbers seem not necessarily significant in terms of improvements, it could lead to more improvements in subsequent works.

**Weaknesses:**

1) The method utilizes a pre-trained SwinIR [1]. However, according to the Table 2 from SwinIR, the performance of SwinIR for the task super-resolution is ~6db higher in terms of PSNR on set5, which is confusing. The authors should include the SwinIR for all of the test datasets as a baseline and explain the reason there is a performance drop after adapting their method.

2) There is another main weakness in the experimental results. There seems to be no discussion on the sources of randomness for the proposed method and the compared works. I would assume for the proposed method, there could be randomness involved in all three main stages: the pre-training, the refinement process, and the Algorithm 1 itself. The same issue could also be true for the considered compared methods. The experimental study should consider multiple runs for each of the mentioned stages and provide individual or combined analysis on the distribution of results rather than just a single run. The reviewers need to be sure whether the provided performance numbers are the worst single runs of compared methods versus the best single run of the proposed method, or are average cases for all considered methods. Without providing such analysis using multiple runs, it is hard to assess the significance of the results for the proposed method.

[1] Liang, J. Cao, G. Sun, K. Zhang, L. Van G., and R. Timofte. Swinir: Image restoration using swin transformer. ICCV, 2021.

**Questions:**

3) The authors should include other evaluation metrics, such as SSIM and LPIPS, for the results. That would help to make sure the performance is not biased towards a single evaluation metric.

4) The authors mention one of the limitations for the proposed method is  “the assumption that the restoration prior used for inference performs MMSE estimation.” However, it is not clear if for the compared methods, they have included any of recent methods that are trained based on l_1 loss and SSIM to clarify how much this limitation affects the final performance in different applications.

5) Using a number of different implicit priors rather than only a single architecture while comparing their performance numbers before and after applying the proposed method would clarify to what extend the method is sensitive to the implicit prior’s performance.

---

> ### Author Response · Authors · 2023-11-19
> **Response to reviewer v5mr**
>
> Thank you for your valuable feedback and positive comments on our work. Please see below for our point-by-point responses to your comments.
>
> >1: The method utilizes a pre-trained SwinIR [1]. However, according to the Table 2 from SwinIR, the performance of SwinIR for the task super-resolution is ~6db higher in terms of PSNR on Set5, which is confusing. The authors should include the SwinIR for all of the test datasets as a baseline and explain the reason there is a performance drop after adapting their method.
>
> We are addressing a different problem than the original SwinIR paper. The original SwinIR paper shows that SwinIR trained for bicubic SR performs well on bicubic SR. Our work shows that SwinIR trained on the bicubic SR can be a good prior to solve **other restoration tasks**. Table 2 in our paper uses SwinIR trained for bicubic SR as a prior to solve what we call the SISR task (which is SR under Gaussian blur kernels and additive white Gaussian noise). Figure 9 in Section B.4 of the supplement shows that SwinIR trained for bicubic SR cannot directly handle SISR, but provides an excellent prior for our DRP method to perform SISR.
>
> >2: There is another main weakness in the experimental results. There seems to be no discussion on the sources of randomness for the proposed method and the compared works. I would assume for the proposed method, there could be randomness involved in all three main stages: the pre-training, the refinement process, and the Algorithm 1 itself. The same issue could also be true for the considered compared methods. The experimental study should consider multiple runs for each of the mentioned stages and provide individual or combined analysis on the distribution of results rather than just a single run. The reviewers need to be sure whether the provided performance numbers are the worst single runs of compared methods versus the best single run of the proposed method, or are average cases for all considered methods. Without providing such analysis using multiple runs, it is hard to assess the significance of the results for the proposed method.
>
> All the experiments presented in the paper use the same random seed to ensure consistency for all methods. We also always use the official models provided on GitHub and follow the recommended settings for each method. Prompted by your comment, we ran new simulations using additional random seeds. Table 14 in Section B.10 of the supplement reports these results. Note how DRP consistently outperforms DPIR for all Seed IDs, which indicates the robustness of our results to multiple random runs.
>
> >3: The authors should include other evaluation metrics, such as SSIM and LPIPS, for the results. That would help to make sure the performance is not biased towards a single evaluation metric.
>
> This is an excellent suggestion. Prompted by your comment, we report SSIM and LPIPS as additional evaluation metrics in Section B.5 of the supplement. Note the excellent performance of DRP for all metrics – PSNR, SSIM, and LPIPS – on all tasks.
>
> >4: The authors mention one of the limitations for the proposed method is  “the assumption that the restoration prior used for inference performs MMSE estimation.” However, it is not clear if for the compared methods, they have included any of recent methods that are trained based on l_1 loss and SSIM to clarify how much this limitation affects the final performance in different applications.
>
> The MMSE assumption is **only** necessary for our theoretical analysis of DRP. DRP can be run using models trained on L1 or SSIM losses. Prompted by your comment, we retrained SwinIR using the same settings introduced in Section 5.1, but using the L1 loss instead of L2. The results are presented in Table 15 in Section B.11 of the supplement. Note how there are no significant differences between DRP using L2 or L1 SwinIR priors, which suggests that numerically DRP is stable with or without the MMSE assumption.
>
> >5: Using a number of different implicit priors rather than only a single architecture while comparing their performance numbers before and after applying the proposed method would clarify to what extend the method is sensitive to the implicit prior’s performance.
>
> Section B.4 in the supplement shows the performance before and after DRP. Note how the SwinIR model trained for bicubic SR struggles on the SISR task (which is SR under Gaussian blur kernels). However, once it is integrated as a prior into DRP, it can achieve the SOTA performance on the new task.
>
> Prompted by your comment, we ran new experiments by including another restoration prior corresponding to a deep unfolding network for solving the image deblurring task. The results are presented in Section B.8 of the supplement. As can be seen in Table 13, the new deblurring prior cannot directly handle the SISR task; yet, it can serve as an excellent prior for regularizing SISR within our DRP method.

---

> ### Author Response · Authors · 2023-11-21
> **Additional response to Reviewer v5mr**
>
> Dear Reviewer v5mr, please let us know if there is something we can do before the deadline to improve your evaluation of our work.
>
> > 5: Using a number of different implicit priors rather than only a single architecture while comparing their performance numbers before and after applying the proposed method would clarify to what extend the method is sensitive to the implicit prior’s performance.
>
> Prompted by the feedback, we have completed an additional experiment using the DRUNet architecture trained for SR as another SR prior for DRP. The new results are presented in Tables 11 and 12 of Section B.8. The new DRP (DRUNet SR) attains comparable performance to DRP (SwinIR SR), consistently outperforming DPIR (DRUNet denoiser). The new results confirm the effectiveness of DRP and provide experimental validation for its robustness across different SR network architectures (both SwinIR and DRUNet).

---

> ### Comment · Reviewer_v5mr · 2023-11-21
> **Response to the authors**
>
> Thanks to the authors for carefully addressing the points in my review.
> After going through the responses and the discussions with other reviewers,
> I can see that most concerns are resolved.
> Thus, I am raising the score.
> For the camera ready version, the authors could apply all evaluation metrics to the results in the main paper as well.

---

> > ### Author Response · Authors · 2023-11-22
> > **Response to Reviewer v5mr**
> >
> > Thank you for your positive feedback and increasing your score. Please let us know if there is anything we can do more.

---

### Author Response · Authors · 2023-11-19
**Response to all reviewers:**

Thank you all for providing us with valuable feedback. We provide detailed answers to all the comments below. To better address some of them, we ran additional simulations, albeit within the time constraints of the limited rebuttal window. Most of these results were included in the dedicated sections of the revised supplementary material. The revised part in the main paper is indicated with red highlighting.

---

### Author Response · Authors · 2023-11-21
**Comment to all reviewers and area chairs**

Dear all, we are nearing the end of the discussion period. We have responded to all the comments with one reviewer raising their initial score so far. We hope that our responses will help other reviewers to also see the value in our work.

We are enthusiastic about this work due to multiple reasons: **(a)** growing interest in using deep models as priors for solving inverse problems in imaging, **(b)** novelty of our principled approach for using general restoration networks as priors, **(c)** novelty of our theoretical analysis that extends the existing theory beyond deep denoisers, **(d)** extensive numerical experiments that provide concrete evidence that our approach can lead to improvements over the state-of-the-art.

Please let us know if there is anything else we can clarify or answer until the deadline.

---

### Meta-Review · Area_Chair_DTzU · 2023-12-01

**Metareview:**

This paper aims to generalize the concept of utilizing pre-trained deep denoisers for plug-and-play image restoration, as there is no existing theory suggesting that denoisers are universally the optimal priors.

The main concern raised by the reviewers pertains to the motivation. Specifically, they highlight that the paper lacks sufficient discussion and analysis on why restoration networks for problems beyond denoiser can yield superior outcomes compared to denoiser. Besides, the proposed method also leads to a new problem of choosing operator H, which seems like it hasn’t been thoroughly discussed in the paper. Therefore, we were actually uncertain about the reasons for introducing H and determining how to select H. Additionally, the AC concurs with reviewer RzPk that there may be doubts about the necessity of this generalization, given that advanced neural network architectures can provide sufficient flexibility for deep denoising priors, i.e., covering the role of H through network design. Besides, selecting H=I should not be the worst choice. It is thus essential to include a discussion about the types of H that may lead to performance degradation.

Anyway, the authors show that after introducing H, the algorithm can still converge under some assumptions, and it can work with the H selected by the authors in some cases. I have no objection to the acceptance of this paper.

**Justification For Why Not Higher Score:**

The paper raises some questions, but many of them are left unresolved.
The methods are relatively incremental in terms of technique.

**Justification For Why Not Lower Score:**

I understand the paper’s contribution

---

### Decision · Program_Chairs · 2024-01-16

Accept (poster)